# The Molecular Profile of Soil Microbial Communities Inhabiting a Cambrian Host Rock

**DOI:** 10.3390/microorganisms12030513

**Published:** 2024-03-02

**Authors:** Ting Huang, Daniel Carrizo, Laura Sánchez-García, Qitao Hu, Angélica Anglés, David Gómez-Ortiz, Liang-Liang Yu, David C. Fernández-Remolar

**Affiliations:** 1SKL Lunar and Planetary Sciences, Macau University of Science and Technology, Macau 999078, China; thuang@must.edu.mo (T.H.);; 2CNSA Macau Center for Space Exploration and Science, Macau 999078, China; 3Centro de Astrobiología (INTA-CSIC), 28850 Madrid, Spain; dcarrizo@cab.inta-csic.es (D.C.); lsanchez@cab.inta-csic.es (L.S.-G.); 4Blue Marble Space Institute of Science, Seattle, WA 98104, USA; angelica.angles@bmsis.org; 5ESCET-Área de Geología, Universidad Rey Juan Carlos, 28933 Móstoles, Spain; david.gomez@urjc.es; 6Institute of Science and Technology for Deep Space Exploration, Nanjing University, Suzhou Campus, Suzhou 215163, China; 7Carl Sagan Center, The SETI Institute, Mountain View, CA 94043, USA

**Keywords:** soil microorganisms, Cambrian host rock, GC-MS, ToF-SIMS, contamination control

## Abstract

The process of soil genesis unfolds as pioneering microbial communities colonize mineral substrates, enriching them with biomolecules released from bedrock. The resultant intricate surface units emerge from a complex interplay among microbiota and plant communities. Under these conditions, host rocks undergo initial weathering through microbial activity, rendering them far from pristine and challenging the quest for biomarkers in ancient sedimentary rocks. In addressing this challenge, a comprehensive analysis utilizing Gas Chromatography Mass Spectrometry (GC-MS) and Time-of-Flight Secondary Ion Mass Spectrometry (ToF-SIMS) was conducted on a 520-Ma-old Cambrian rock. This investigation revealed a diverse molecular assemblage with comprising alkanols, sterols, fatty acids, glycerolipids, wax esters, and nitrogen-bearing compounds. Notably, elevated levels of bacterial C_16_, C_18_ and C_14_ fatty acids, iso and anteiso methyl-branched fatty acids, as well as fungal sterols, long-chained fatty acids, and alcohols, consistently align with a consortium of bacteria and fungi accessing complex organic matter within a soil-type ecosystem. The prominence of bacterial and fungal lipids alongside maturity indicators denotes derivation from heterotrophic activity rather than ancient preservation or marine sources. Moreover, the identification of long-chain (>C22) n-alkanols, even-carbon-numbered long chain (>C20) fatty acids, and campesterol, as well as stigmastanol, provides confirmation of plant residue inputs. Furthermore, findings highlight the ability of contemporary soil microbiota to inhabit rocky substrates actively, requiring strict contamination controls when evaluating ancient molecular biosignatures or extraterrestrial materials collected.

## 1. Introduction

Soil genesis depends profoundly on the establishment of lithobiontic microbial consortiums that actively mine growth-limiting mineral nutrients through chemical and physical weathering reactions at bedrock boundaries. These early heterotrophic colonists—encompassing bacteria, fungi, algae and lichens—propagate the initial conditions, enabling more complex ecosystems via generation of bioavailable elements and incremental fertility enrichment of the substrate [1]. Microbes are drawn to freshly exposed bedrock to obtain key macronutrients like iron, potassium, magnesium, and phosphorus entrapped within crystalline structures of primary minerals, in particular targeting more rapidly weathering igneous species [2]. They accelerate decomposition through production of metal-chelating organic acids and ligands that solubilize and release critical elements from mineral matrices orders of magnitude faster than background geochemical processes alone.

Clay minerals are the most abundant mineral type in soils [3], and have close interactions with microbes, playing critical roles in soil processes [4]. Microbially mediated clay mineral formations, like kaolinite, occur via precipitation of aluminate gel and organic products [5]. Microbes can enhance clay weathering, altering properties like cation exchange capacity and water holding capacity [6]. During the microbial weathering process, cations released by clay minerals are an essential nutrient source for plants [7]. They also promote the formation of other minerals, and influence their composition [8]. For example, the interaction between *Rhodopseudomonas* sp. and kaolinite leads to the transformation of kaolinite to gibbsite [9]; Montmorillonite interacts with *Paenibacillus* sp. and releases Ca^2+^, which then combines with carbonate ions produced through bacterial metabolism to form calcite [10].

As initial microbial colonists decompose host materials through metabolic activity and biosynthesis, the physiochemistry of the substrate progressively shifts from properties of the underlying lithic parent material toward those representing a nascent soil habitat. Microbially produced organic acids and biofilm physically disrupt bedrock, increasing porosity and generating organo-mineral particles cemented by microbial cell debris and extracellular polymeric substances [11]. The increase in porosity makes water more accessible which, in turn, facilitates microbial colonization within the rocks [12]. Furthermore, redox gradients manifest proximal to respiring microbial cells, as production of extracellular metabolites and cell envelope biochemical transformations locally toggle reduction–oxidation states relative to the bulk mineral phase. By extracting nutrients, microbes further transform bulk elemental ratios and phase-partitioning of compounds from rock-dominated distributions toward biological preference [13].

The pioneering lithobiontic consortiums adept at liberating essential growth factors from recalcitrant minerals. Primary colonists comprise stress-tolerant heterotrophs often displaying lithoautotrophic capacity using different ions as electron donors and acceptors [14]. Fungi further recycle organic compounds and augment initial acidification reactions through select metal-chelating organic acids, occurring under a strong association with the bacterial communities [15,16]. As the cycling of elements progresses, it facilitates the emergence of secondary organisms, notably the Rhizobiales group, fostering the diversification of plant and invertebrate communities [17], crucial for the efficient input of carbon flows through the cycling of litter.

The investigation of soil microbiota inhabiting lithological materials, the progenitors of soils, has advanced through the application of microbiological and molecular biology techniques [17,18]. However, there have been limited efforts to systematically monitor the microbial influence on the mineral matrix of the host rock by integrating spectral techniques with mapping capabilities. In this context, a comprehensive molecular analysis of a 520-million-year-old Cambrian host rock from the Sierra de Córdoba in South Spain was conducted. The host rock formed under a sharp rise greenhouse [19]. If ancient biomolecules from that time were well preserved in the rocks without subsequent alterations, this would provide important clues to study the ancient communities during the transition between the Ediacaran and Cambrian. However, the preservation of molecular composition is indicative of underground microbial colonization processes. It aligns with the soil formation to some extent, and the concentrations, diversity, and association with the mineral matrix suggest that these compounds are recent or exceptionally well-preserved. Considering this, it is essential to conduct a detailed investigation into the interaction between modern soil microbes and ancient host rocks.

The molecular characterization was conducted through a dual approach involving Gas Chromatography Mass Spectrometry (GC-MS) compositional profiling and Time-of-Flight Secondary Ion Mass Spectrometry (ToF-SIMS) microscale mapping. This combined analytical strategy enhances our understanding of the molecular distribution within the rock matrix, providing valuable insights into the intricate relationship between the detected compounds and the geological substrate. The high-resolution imaging capabilities of ToF-SIMS contribute to the maximization of information pertaining to the spatial distribution of these molecules within the rock matrix, enriching our comprehension of their contextual significance.

## 2. Materials and Methods

### 2.1. Geologic Background and Sampling

The samples analyzed for organics come from early Cambrian sedimentary outcrops in the Las Ermitas location of the Sierra de Córdoba (37°55′08″ N, 4°49′36″ W), occurring at the North of the Córdoba city, South Spain (Figure 1A,C). The Sierra de Córdoba is emplaced on the western margin of the Córdoba-Alanis Domain in the Ossa-Morena tectonostratigraphic zone of the Iberian Massif [20,21]. Such a domain records the Neoproterozoic to Paleozoic evolution resulting from the interaction among the Gondwana, Laurentia, Baltica, and Iberia continental units in two consecutive orogenetic cycles, the Cadomian and the Hercynian [22].

The stratigraphic record of such a long-term geodynamic evolution in the Sierra de Córdoba has resulted on the definition of five different units, which sedimentary and paleontological content have been the foundations to build the chronostratigraphic scale going from the late Neoproterozoic to the lower Cambrian in Iberia [23,24,25,26]. The lithological and chronostratigraphic units (Figure 1B) start with the San Jerónimo Fm defined from the volcanosedimentary materials likely of the terminal Ediacaran to the early Terrenuvian age [25]. It is followed by the Torreárboles Fm, consisting of conglomerates, sandstones and siltstones, with strong lateral facies and thickness lying unconformably on the volcanosedimentary deposits [27]. The Torreárboles detrital sequence gradually varies upwards, ending in a carbonate and siliciclastic mixed unit of the Pedroche Fm [23,28,29,30] containing alternating reefal mounds, nodulose limestones, sandstones, and siltstones, which locally contain phosphoritic levels as in the Las Ermitas location [31]. Such a unit gradually varies upwards to the Santo Domingo Fm, consisting of alternating dolomites, silicified carbonates, and siltstones. The Cambrian sedimentary record is ended by the Los Villares Fm through the occurrence of a finning upwards sequence of quartzites, sandstones, and siltstones. While the San Domingo Fm is lacking a fossil record, Los Villares Fm contains an abundant record of trilobites that have been the base to define the Bilbilian and Caesaraugustian Iberian stages in the region [27].

The focus of molecular analysis in this study centered on samples obtained from the Pedroche Formation, situated in unconformable juxtaposition with the San Jerónimo Formation (Figure 1C and Figure 2A–C). Specifically, the sampling targeted carbonatic and green siltstones harboring phosphatic compounds, situated beneath a soil cover ranging from approximately 30 to 50 cm. This soil cover delineates a somewhat immature pedogenetic unit reminiscent of a leptosol, predominantly found in the steep and mountainous sectors of the Sierra Morena [32,33]. Notably, the specific characteristics of these leptosols exhibit variability in argillaceous horizons, influenced by the evolving geological settings. Of particular interest in an evolutionary context is the discernible transition of such soils from skeletal formations since the Pleistocene [32]. This evolution warrants careful scrutiny, as it contributes to our understanding of the formation processes underlying the molecular associations identified in the host rock.

### 2.2. Methods

Sample cores from the inner part of the collected rocks were extracted using a rock saw. Then, the samples were prepared for the CG-MS and ToF-SIMS analyses. First, sample 95E5/2 was cleaned for 15 min using an ultrasonic bath inside a glass flask using a dichloromethane/methanol 3:1 solution. Later on, the sample was dried in a furnace at 50 °C, and was ground before proceeding with the organic extraction.

Polished sections for sample 95E5/3 were prepared for mineral and textural analysis under a microscope and molecular surface analysis by ToF-SIMS [34]. The surface sample was extra-polished using a 0.3 mm alumina paste to reduce surface imperfections that produce frequent interferences during the surface analysis.

#### 2.2.1. Organic Extraction, Fractionation, and Analysis under GC-MS

About 130 g dry weights of the powdered samples were Soxhlet extracted with organic solvents (dichloromethane and methanol, 3:1 vol.) and fractionated as described elsewhere [35]. A mixture of internal standards (tetracosane-D50, myristic acid-D27, and 2-hexadecanol) was added prior to the extraction for quantification (recovery of 77 ± 13%). Procedural blanks were performed throughout the entire process to confirm that the compounds identified were indigenous to the samples. For analysis, the polar fraction containing alcohols was also derivatized using BSTFA (Sigma Aldrich, Madrid, Spain) at 80 °C for 60 min prior to injection in the GC-MS. Thus, acids and alcohols were detected as fatty acid methyl esters (FAME) and trimethylsilyl (TMS) derivates, respectively.

Organic compounds were analyzed using GC-MS for identification and quantification, and GC-IRMS for compound-specific stable-carbon isotopic analysis. For GC-MS analysis, we used a gas chromatography system (8860 GC) coupled to a mass spectrometer (5977B MSD) (Agilent Technologies, Santa Clara, CA, USA), operating with electron ionization at 70 eV and scanning from *m*/*z* 50 to 650 (analytical details in [35]). Compound identification was based on retention time and mass spectra comparison with reference materials and the NIST mass spectral database. Quantification was performed with the use of external standards of, FAMEs (C_6_ to C_24_) and *n*-alkanols (C_14_, C_18_ and C_22_), all supplied by Sigma-Aldrich (Madrid, Spain). The stable-carbon isotopic composition was measured using a Trace GC 1310 ultra and ISQ QD MS interfaced to a MAT 253 IRMS, Thermo Fisher Scientific (Waltham, MA, USA). The analytical conditions for the GC-IRMS are detailed elsewhere [36]. For fatty acids and sterols, the carbon isotope ratios were corrected for the carbon added during methylation and silylation, respectively [36]. The δ^13^C ratio (‰) was expressed relative to the Pee Dee Belemnite (PDB) standard. The accuracy of the carbon isotopic ratio was tested by running reference mixtures (Indiana University, USA) of known isotopic compositions of *n*-alkanes (A6) and FAMEs (F8) in every three samples. 

#### 2.2.2. ToF-SIMS Analysis

The ToF-SIMS technique establishes a direct association between the inorganic and organic compositions of the sample microstructure. Before analysis, the thin sections were cleaned by surface sputtering with a 100 nA 500 3 V oxygen ion beam for 3 s on a square of 500 × 500 µm. The ToF-SIMS analyses were performed using a ToF-SIMS IV (ION-TOF, Münster, Germany) operated at a pressure of 5 × 10^−9^ mbar. Samples were bombarded with a pulsed Bismuth liquid metal ion source (Bi_3_^+^) at 25 keV. The gun was operated with a 20 ns pulse width, 0.3 pA pulsed ion current, for a dosage lower than 5 × 10^11^ ions/cm^2^, well below the threshold level of 1 × 10^13^ ions/cm^2^ generally accepted for static SIMS conditions. Secondary ions were detected with a reflector time-of-flight analyzer, a multichannel plate (MCPs), and a time-to-digital converter (TDC). Charge neutralization was achieved with a low energy (20 eV) electron flood gun. Secondary ion spectra were acquired from a randomly rasterized surface area of 500 μm × 500 μm within the sample’s surface. Secondary ions were extracted with 2 kV accelerating voltage, and were post-accelerated to 10 keV kinetic energy just before hitting the detector. Mass spectral acquisition and image analysis were obtained within the ION-TOF Ion Spec and Ion image software (version 6.8). Each ion image shown is normalized to the intensity of the brightest pixel. This intensity value is assigned to the color value of 256. Zero intensity is assigned to the color value 0. All other intensities are assigned accordingly, using a linear relationship. The target areas for analysis were previously selected by using the visible light and the Scanning Electron Microscope (SEM) integrated into the ToF-SIMS suite.

Furthermore, a multivariate analysis combining the spectral and spatial distribution in the sample was run to find correlations between fragment occurrence and molecular fragmentation patterns. Mass spectral acquisition and image analysis were performed within the ION-TOF Surface Lab software suit (version 7.0), including the overload of Red, Green, and Blue (RGB) images for molecular mapping [34,37]. The analysis of the molecular fragmentation pattern was completed through the Chemical online tool [38] and the open source mass spectrometry tool mMass [39]. ChemSpider [40], METLIN [41], and LIPID MAPS structure databases [40,41,42,43] were used as sources of information to identify molecular fragments and compounds.

## 3. Results

### 3.1. GC-MS Results

The organic extraction of sample 95E5/2 collected in the carbonatic deposits (Figure 2A,B) yielded compounds of different configurations (e.g., linear and saturated, branched and unsaturated, acyclic, cyclic, etc.). They were from acidic (fatty acids), and polar (alcohols) fractions. The *n*-fatty acids (Figure 3A; Appendix A) show a predominance of low molecular weight (LMW) congeners (≤*n*-C_20_) of even-over-odd preference, together showing a total concentration of 4.04 µg·gdw^−1^. The most abundant *n*-fatty acid was *n*-C_16_ (2.23 µg·gdw^−1^), followed by *n*-C_18_ (1.12 µg·gdw^−1^). Other secondary peaks were those of the high molecular weight (HMW) congeners *n*-C_24_, *n*-C_22_ and *n*-C_26_ (Figure 3A).

The compositional profile of branched fatty acids (Figure 3B; Appendix A) within the sample exhibited a spectrum ranging from dodecanoic acid to hexacosanoic acid derivatives, wherein methyl branches adorned the carbon backbone. Notably, both iso and anteiso methyl branching configurations manifested throughout this extensive chain length continuum. The quantification of these methyl-branched compounds revealed concentrations spanning from 0.005 to 0.063 µg·gdw^−1^. Noteworthy is the occurrence of odd- and even-numbered homologs, each featuring methyl groups at either the iso or anteiso positions, pervasive among the majority of observed chain lengths, whose distribution attained its zenith at C_15_ and C_17_ branched chain compounds. Supplementary to the fatty acid spectrum, the initial milieu encompassed distinctive compounds such as 10-methyl undecanoic acid (0.002 µg·gdw^−1^) and 4-methyl dodecanoic acid (0.011 µg·gdw^−1^), alongside mono-unsaturated counterparts exemplified by (Z)-11-tetradecenoic acid (0.008 µg·gdw^−1^). Mid-chain methyl branching phenomena were observed on diverse straight-chain fatty acids, notably 13-methyl (0.043 µg·gdw^−1^) and 14-methyl (0.010 µg·gdw^−1^) pentadecanoic acids, and 10-methyl hexadecanoic acid (0.046 µg·gdw^−1^).

Concurrently, an assemblage of octadecanoic acid isomers contributed to the long unsaturated acids profile, with 9,12-octadecadienoic (0.109 µg·gdw^−1^), 9-octadecenoic (0.584 µg·gdw^−1^), and 11-octadecenoic acid (0.23 µg·gdw^−1^) standing out prominently. Additionally, dicarboxylic fatty acids made their presence felt, featuring hexanedioic acid (0.003 µg·gdw^−1^), octanedioic acid (0.004 µg·gdw^−1^), and docosanedioic acid (0.014 µg·gdw^−1^).

In turn, the distribution of *n*-alkanols does not have a clear predominance of HMW or LMW compounds, but a prevalence of even-over-odd chains (Figure 4A; Appendix A). The *n*-alkanols series (C_12_ to C_28_) shows a concentration (1.06 µg·gdw^−1^) four times lower than that of *n*-fatty acids (4.78 µg·gdw^−1^). In addition to the *n*-alkanols, the polar fraction also contained a set of C_27_ to C_29_ sterols (Figure 4B; Appendix A) which, together, showed a total concentration of 0.39 µg·gdw^−1^. The steroids series included a few sterols (cholesterol and campesterol), and different stanols and other sterol derivatives (cholestadienol, ergostadienol, stigmastene, and stigmastanol) (Appendix A).

Different proxies calculated on the distribution of *n*-fatty acids (Appendix A) were used to assess their sources. First, an average carbon length (ACL) of 18 points to microbial sources [44]. On the other hand, a value of 3.7 for the carbon preference index (CPI) of the *n*-fatty acids (C_11_ to C_28_) is consistent with a more extant than extinct biomass [45,46]. Furthermore, relatively higher values of the Low-to-High Molecular Weight (LMW/HMW) ratio in the *n*-fatty acids (5.5) supported a prevalent proportion of prokaryotic over eukaryotic biomass [47,48].

The compound-specific isotopic analysis (CSIA) of *n*-fatty acids shows a δ^13^C ranging from −26.5 to −36.6‰ (Figure 5), where the average was −28.7‰ for the LMW (≤*n*-C_21_) and −33.2‰ for the HMW (≥*n*-C_22_).

### 3.2. ToF-SIMS Results

The molecular analysis under ToF-SIMS extensively inspected the sample 95E5/3 in two different target areas 1 and 2 (TA1 and TA2) for positive and negative ions, respectively. In addition, the analysis of one additional target area 3 (TA3) for positive ions was performed to support the results obtained in TA1 and TA2 (Figure 6, Figure 7 and Figure 8).

#### 3.2.1. ToF-SIMS Molecular Mapping

***Mapping of positive and negative inorganic ions***. The distribution of inorganic and organic ions was performed to understand the association among the inorganic and organic compounds with the different elements of the structure and texture sample. In sample 95E5/3, the ion distribution characterized by the ToF-SIMS mapping capability greatly depends on the arrangement of the rock’s primary components like skeletal/non-skeletal grains and bioclasts, and the mineral matrix or cement [34,49].

In this regard, the distributions of inorganic ions in TA1 (Figure 7) show an intensity (I) >500 counts per second (cps), including K^+^ (*m*/*z* 38.97), Na^+^ (*m*/*z* 22.99), Na_2_OH^+^ (*m*/*z* 62.98), Ca^+^ (*m*/*z* 33.96), CaOH^+^ (*m*/*z* 56.97), Ca_2_O_2_H^+^ (*m*/*z* 112.92), P_2_H_3_^+^ (*m*/*z* 64.97), Ca_3_PO_5_^+^ (*m*/*z* 230.83), KNaOH^+^ (*m*/*z* 78.95), Na_2_Cl^+^ (*m*/*z* 80.94), K_2_Cl^+^ (*m*/*z* 112.9), NaKCl^+^ (*m*/*z* 96.92), MgCaCl^+^ (*m*/*z* 98.92), Ca_2_Cl^+^ (*m*/*z* 114.89), and NaKMgH_2_^+^ (*m*/*z* 87.95), following the sample fabric and texture (Figure 7A–D). The ToF-SIMS TA1 mapping shows a close relation among P- and Ca-bearing ion complexes (e.g., CaOH^+^, Ca_2_O_2_H^+^, and Ca_3_PO_5_^+^), which correspond with the occurrence of carbonate fluorapatite (Figure 7B). Interestingly, the more homogeneous distribution of Ca^+^ suggests that it results from the carbonate and fluorapatite combined occurrence (Figure 7A–C). A second group of Na- and Cl-bearing complex ions having K, Ca, Mg, and Fe are likely correlated with the presence of silica and the carbonate matrix depleted in organics (Figure 7C; Appendix A). In turn, Fe, Mg, and Al intensity sum correlates well with the Si-bearing positive ions like Si^+^ (*m*/*z* 27.97), SiH^+^ (*m*/*z* 28.98), SiH_3_O^+^ (*m*/*z* 47.00), and Si_3_HO^+^ (*m*/*z* 100.94), which show a homogeneous distribution with a higher concentration increase at the external part of the phosphatic area (Figure 7C). Such silica distribution agrees with the SEM and microscope observations, where a quartz and chlorite intermix inside the apatite steinkerns is followed by an enrichment in silica minerals in the external side of the phosphatic casts [50].

In turn, the ToF-SIMS mapping of TA1 positive ions shows that the lipid distribution has a particular complementary occurrence, which occupies the areas where apatite and phyllosilicates are less present (Figure 7B,C). However, the ToF-SIMS results also show that organics can occur associated with silica and phosphate at a lower concentration. This suggests that the lipidic component is mainly associated with the carbonate fraction, but occurrence is locally linked with the mineralization of phosphate and silica in varying microenvironmental and microbial preserving conditions.

Furthermore, a couple of positive ions, like Ti^+^, Pb^+^ and NH_4_^+^, have a particular distribution differing from the ionic complexes described above (Figure 7D). The Ti^+^ cation occurs as 1- to 20-micron-sized multiple elements, which are uncorrelated with the distribution of any other cation (Figure 7D). Such an occurrence suggests that they correspond with detrital particles integrated into the phosphatic and carbonatic matrix. The distribution of total Pb^+^ (including ^204^Pb^+^, ^206^Pb^+^, ^207^Pb^+^, and ^208^Pb^+^) is very intriguing, as it follows the distribution of phosphatic and carbonatic matrix (Figure 7D), but it is lacking in the chlorite area traced by the Cl- and Na-bearing cationic complexes. Although the Pb^+^ presence could be solely attributed to sample contamination, its correlation with the carbonatic and phosphatic minerals suggests that it formed as any element released to the lagoon from weathering, though hydrothermal sources might not be discarded [51]. Furthermore, the NH_4_^+^ distribution follows the occurrence of the N-bearing organic compounds like trialkylamines, which are equally found in the carbonatic and phosphatic fractions, as will be discussed below.

The distribution of negative ions was examined in TA2 to aggregate information regarding the occurrence of inorganic compounds in the sample (Figure 8). As a result, different anionic complexes like PO_2_^−^ (*m*/*z* 62.96), PO_3_^−^ (*m*/*z* 78.96), CO_3_^−^ (*m*/*z* 59.98), S^−^ (*m*/*z* 31.97), HS^−^ (*m*/*z* 32.98), SO_3_^−^ (*m*/*z* 76.96), SO_4_^−^ (*m*/*z* 95.95), NaH_2_S_3_O_2_^−^ (*m*/*z* 152.91), SiO_3_H^−^ (*m*/*z* 76.97), Si_2_HO_4_^−^ (*m*/*z* 120.94), NO_2_^−^ (*m*/*z* 45.99), and NO_3_^−^ (*m*/*z* 61.99) were found. Their distribution greatly follows the rock microstructure constrained by the phosphate, silica, and carbonate composition (Figure 8A–D), as seen in the presence of the positive cations above. In this regard, the boundary among the phosphatic steinkern and the carbonatic matrix recognized by the ToF-SIMS visible camera and SEM (Figure 8A) is clearly shown by a sharp decrease in PO_2_^−^ and PO_3_^−^ concentration when in contact with the lipid-rich area, defined by the fragment C_3_H_3_O_2_^−^ (*m*/*z* 71.01) (Figure 8B). Interestingly, the lipidic fraction, yet associated with the calcitic matrix, also appears in the phosphatic matrix, likely correlated with more carbonatic areas inside the apatite frame. As the lipidic fraction shows fabric continuity in the calcitic and apatitic areas, it could be that the TA2 is showing a mineralized reaction in front of a calcite replacement by phosphate, where the organic compounds have been preserved. Such a particular lipidic fabric could also result from an organic infilling produced by microbial boring crosscutting the mineral interphase.

The fabric of the Si-bearing negative ions (e.g., SiO_3_H^−^, Si_2_HO_4_^−^) is characterized by mineralized elements of varying sizes (10–200 microns) and shapes, which appear in the phosphatic and carbonatic matrix (Figure 8A–C). While the silica anions counteract the distribution of the other anions such as P-bearing, they also partially combine with the lipidic and phosphatic fraction. Indeed, the Si-bearing anions show a relatively high concentration in the phosphatic matrix, which in the carbonatic matrix is only shown in the form of particulate silica (Figure 8B,C). Moreover, the appearance of the N- and Cl-anions (Figure 8D,E) correlates with the Si- and S-bearing negative complexes, which suggests they come from an intermixing among different mineral phases and the lipidic fraction. Furthermore, the SO_x_^−^_(x = 2–4)_ ions (Figure 8E) show a higher concentration in the phosphatic and lipidic areas, but dissimilate the areas with HS^−^ and S^−^, and are counterparts to the occurrence of the Si^−^ (Figure 8B). Such a configuration should be related to sulfate formation linked to the fluid percolation, and the sulfide oxidation of the host rock containing pyrite [52]. Interestingly, bromine and fluorine appear uncorrelated with the distribution of the Cl-bearing anions (Appendix A), suggesting that the halides have different sources. Additionally, a high-intensity peak (I >1500 cps) at 89, which aligns well with C_2_HO_4_^−^, indicates the presence of oxalic acid.

***Molecular mapping of organic anions***. The ToF-SIMS mapping capabilities have also unlocked the distribution of the main organic categories in the sample, which are mostly represented by different lipid groups (Figure 9). Indeed, the ToF-SIMS analysis allows us to perform a simple compound separation based on the compound distribution in the sample. In this regard, the different organic groups can eventually follow different textural patterns, which are related to the compound sources [34]. By associating all the imaging data obtained by the ToF-SIMS including ion mapping, visible, and SEM images, it is possible to identify several morphologic groups through the distribution of the organic compounds in TA1 and TA2 (Figure 9; Appendix A).

**Group 1 (G1).** It consists of an averaged 150-micron thick high-intensity band characterized by peaks of major positive ions at 313.29 (C_19_H_37_O_3_^+^), 327.29 (C_20_H_39_O_3_^+^), 341.31 (C_21_H_41_O_3_^+^), 385.35 (C_27_H_45_O_3_^+^), 414.37 (C_29_H_50_O^+^), 523.48 (C_33_H_63_O_4_^+^), 537.50 (C_34_H_65_O_4_^+^), 551.51 (C_35_H_67_O_4_^+^), 565.53 (C_36_H_69_O_4_^+^), and 579.54 (C_37_H_71_O_4_^+^), corresponding to different lipids like monoacyl- and diacylglycerides and steroids (Figure 9A; Appendix A). Such a band goes obliquely from the upper right corner under the phosphatic regions traced by Ca_3_PO_5_^+^ and CaOH^+^ ions (Figure 7B), reaching the TA1 lower axis at 90 microns from the lower left corner. At the middle part of the path, the band splits into four different 80-micron parts with ovoidal to circular morphology, separated by a void area depleted in lipids, which is enriched Cl- and Na-bearing cations forming Group 2 (Figure 7B and Figure 9B). Such a lipid depletion area is also observed in TA3 (Appendix A), suggesting that it corresponds with the carbonatic mineral matrix containing organics and the phosphatic elements. G1 also complements the distribution of Group 3 (G3), which corresponds to a secondary population.

**Group 2 (G2).** Characterized by a maximal intensity for the Cl- and Na-bearing positive complexes (Figure 9B; Appendix A) like Na_2_Cl^+^ (*m*/*z* 80.94), K_2_Cl^+^ (*m*/*z* 112.9), NaKCl^+^ (*m*/*z* 96.92), MgCaCl^+^ (*m*/*z* 98.92), and Ca_2_Cl^+^ (114.89), where G1 defined by the most of lipidic fraction show depletion.

**Group 3 (G3).** Defined by the presence of three major positive ions of wax esters (Figure 9C; Appendix A) like C_14_H_29_O_2_^+^ (*m*/*z* 229.22), C_15_H_31_O_2_^+^ (*m*/*z* 243.22), and C_16_H_33_O_2_^+^ (*m*/*z* 257.25) that appear to form 10 to 80 micron-sized clusters alternating with the lipid distribution defining G1 (Figure 9A; Appendix A). Interestingly, G3 partially occurs at a lower intensity in the areas of maximal intensity for G1 and G2, suggesting that the C_14_ to C_15_ wax esters have two different sources, one of them associated with the occurrence of the glycerolipidic fraction.

**Group 4 (G4).** Identified by a quasi-homogeneous distribution of different N-bearing organics like C_3_H_8_N^+^ (*m*/*z* 58.07), C_6_H_6_NO^+^ (*m*/*z* 60.05), C_24_H_50_N^+^ (*m*/*z* 352.39), C_34_H_72_N^+^ (*m*/*z* 494.56), C_35_H_72_N^+^ (*m*/*z* 506.56), C_35_H_74_N^+^ (*m*/*z* 508.57), C_36_H_76_N^+^ (*m*/*z* 522.59), and C_35_H_72_N^+^ (*m*/*z* 550.62), likely corresponding with trialkylamines, which show an intensity decrease in the external area of the phosphatic elements (Figure 9D). Interestingly, the compound appears an internal texture in the form of <40 micron-sized nodular particles, which meets some rock textural elements like silica-rich. Furthermore, the N-bearing compounds display a higher concentration at the distribution pattern defining G1 (Figure 9A,D), where sterols and glycerolipids have the highest intensity.

#### 3.2.2. ToF-SIMS Spectral Analysis: Compound Classification

Five main groups of organic compounds have been found by analyzing the fragmentation pattern derived from the ToF-SIMS spectrograms collected in the different target areas such as TA1, TA2 and TA3. Such groups are described in the following.

**Fatty acids (FA)**. ToF-SIMS spectral analysis of sample 95E5/3 shows the presence of numerous strong peaks in the positive and negative spectrograms corresponding to the fragmentation of saturated and monounsaturated FAs (Figure 10A,B; Appendix A). In this regard, positive *m*/*z* peaks ranging from 83.05 to 479.52 Da like, e.g., C_14_H_27_O^+^ (*m*/*z* 211.22), C_16_H_29_O^+^ (*m*/*z* 235.22), and C_33_H_67_O^+^ (*m*/*z* 479.52), which agree with the formation of [M + H − H_2_O]^+^ cations (see Appendix A). On the other hand, FAs are also detected in the negative *m*/*z* spectrogram (Appendix A), where they may occur as large [M − H]^−^ peaks going from 157.12 to 423.42 Da, agreeing with C_14_H_27_O_2_^−^ (*m*/*z* 227.0), C_15_H_29_O_2_^−^ (*m*/*z* 241.22), C_16_H_29_O_2_^−^ (*m*/*z* 253.22), C_16_H_31_O_2_^−^ (*m*/*z* 255.23), and C_18_H_35_O_2_^−^ (*m*/*z* 283.26). The minor peaks of negative fatty acid spectra occur at *m*/*z* 209.19, 251.21, 275.20, 277.21, 279.23, and 307.27, fitting C_13_H_21_O_2_^−^, C_16_H_27_O_2_^−^, C_18_H_27_O_2_^−^, C_18_H_29_O_2_^−^, C_18_H_31_O_2_^−^ and C_20_H_35_O_2_^−^ (Appendix A), which correspond with the [M − H]^−^ ions of C_13:2_, C_16:2_, C_18:4_, C_18:3_, C_18:2_, and C_20:2_ polyunsaturated fatty acids (PUFAs).

**Glycerolipids and related compounds.** The analysis of the positive spectral data has led to the detection of a very diverse set of cations that agree well with the fragmentation of mono- and diglycerides (Figure 10A,C and Figure 11A,B; Appendix A) performed through ToF-SIMS [53]. Particularly, a series of strong peaks found at *m/z* 495.45, 523.48, 537.50, 551.51, 565.53, 577.53, 579.54, and 593.57 greatly match C_31_H_59_O_4_^+^, C_33_H_63_O_4_^+^, C_34_H_65_O_4_^+^, C_35_H_67_O_4_^+^, C_36_H_69_O_4_^+^, C_37_H_69_O_4_^+^, C_37_H_71_O_4_^+^, and C_38_H_73_O_4_^+^. Such cations can be associated with the formation of [M + H − H_2_O]^+^ fragments, where M corresponds with DG (14:0/14:0), DG (16:0/14:0), DG (16:0/15:0), DG (18:0/14:0), DG (18:0/15:0), DG (18:1/16:0), DG (18:0/16:0), and DG (18:0/17:0), respectively (Figure 10A,B, Figure 11A,B and Figure 12A). The presence of such a diverse set of diglycerides is supported by the detection of shorter fragments resulting from splitting the compounds at the both sides of C-O ester bond, resulting in diverse fragments (Figure 11A; Appendix A) containing one to three oxygen atoms [53]. It is supported by the observation of positive *m*/*z* peaks like 211.21, 239.25, 241.22, 265.26, 267.28, 283.27, 299.27, 313.29, 341.30, and 371.36, which fit well with cations C_14_H_27_O^+^, C_16_H_31_O^+^, C_18_H_33_O^+^, C_15_H_29_O_2_^+^, C_18_H_35_O^+^, C_18_H_35_O_2_^+^, C_18_H_35_O_3_^+^, C_19_H_37_O_3_^+^, C_21_H_41_O_3_^+^, and C_23_H_47_O_3_^+^, respectively. The formation of the [M − H]^−^ FA ions is also associated directly with the fragmentation process in the glycerides (Figure 10A–C), which is supported by sharing the same compound distribution with the different diglyceride cations [37]. Furthermore, it is likely that the formation of the acylium cations (R-CO^+^) come from the diacylglyceride fragmentation and the loss of H^+^ from detached FAs through the ToF-SIMS sample analysis or by natural degradation processes, which will be further discussed in the following section. Finally, the formation of the [M − H]^−^ anions might be also related to the presence of longer fragments found at *m*/*z* 763.69, 777.69,787.77, 791.69, 803.51, and 805.47 fitting C_48_H_91_O_6_^−^, C_49_H_93_O_6_^−^, C_52_H_94_O_4_^−^, C_50_H_95_O_6_^−^, C_49_H_71_O_9_^−^, and C_44_H_69_O_13_^−^ (Figure 12B) corresponding with different triacylglycerol fragments and likely different glycosylglycerols [54]. FA distribution is highly correlated with the occurrence of triacylglycerols [M − H]^−^ (Figure 12B), which strongly suggests a common origin, as shown by the positive adducts of diacylglycerides and FAs.

**Sterols.** The spectral analysis has yielded valuable insights into the characterization of various *m*/*z* positive peaks (Appendix A), previously identified as originating from the fragmentation of sterols and hopanoids [55,56]. Generally, the smaller fragments (<300 Da) arise from the fragmentation of both sterols and hopanoids (Appendix A), whereas those with *m*/*z* > 350 provide specific signatures of individual sterols (Figure 13) through the identification of [M + H − H_2_O]^+^, [M − H]^+^, and [M]^+^ cations (Appendix A). Notably, the primary sterols found in the sample include cholestadienol (C_27_H_44_O), cholestadiene (C_27_H_44_), cholesterol (C_27_H_46_O), ergostadienol (C_28_H_46_O), ergostenol (C_28_H_44_O), and stigmasterol (C_29_H_50_O). Furthermore, certain peaks may correspond to the presence of steranes and sterenes, such as cholestadiene (C_27_H_44_), ergostene (C_28_H_48_), stigmastene (C_29_H_50_), and stigmastane (C_29_H_52_).

Moreover, the ToF-SIMS spectral data provide indications of the presence of hopanoids (Appendix A), as evidenced by a few <250 Da cations, including C_13_H_21_^+^ (*m*/*z* 177.16), C_14_H_23_^+^ (*m*/*z* 191.18), and C_15_H_25_^+^ (*m*/*z* 205.19). However, it is important to note that these data alone do not furnish sufficient information to identify specific compounds, as achieved through more comprehensive techniques like GC-MS analysis. Additionally, certain larger fragments attributed to steroids are also observed, which can be derived from hopanoids. Examples include C_27_H_43_^+^ and C_27_H_45_^+^, corresponding to hopenes and norhopanes [57]. Nonetheless, no other significant hopanoid fragments are identified, underscoring their comparatively lower concentration in the rock when compared to sterols, as confirmed by the GC-MS analysis. The sterols, on the other hand, exhibit a minimum concentration threshold necessary for their detection by ToF-SIMS.

**Wax esters.** The ToF-SIMS spectral analysis of sample 95E5/3 reveals the presence of several prominent *m*/*z* positive peaks at 229.22, 243.22, and 257.25, exhibiting a distinct distribution pattern (Figure 9C) that differs from the distribution of other lipids in Figure 9A. These peaks, which correspond to C_14_H_29_O_2_^+^, C_15_H_31_O_2_^+^, and C_16_H_33_O_2_^+^ ions, align well with the presence of wax esters [58,59]. The composition and distribution of these wax esters in the sample suggest an origin separate from the source of glycerides and steroids. Additionally, there are other *m*/*z* peaks such as 521.53 (C_35_H_69_O_2_^+^) and 535.53 (C_36_H_71_O_2_^+^) that, although partially obscured by stronger peaks, likely correspond to the presence of larger wax ester chains [58].

**N-bearing compounds: amine derivative, tetraalkylammonium adducts and sphingolipids**. Amine derivatives and tetraalkylammonium adducts are prominent in the observed spectra, as supported by the strong peaks detected at *m*/*z* 18.03, 26.01, and 42.00, corresponding to NH4^+^, CN^−^, and CNO^−^, respectively. These findings provide evidence for the presence of various nitrogen-bearing compounds. Specifically, the detection of different (even positive) ions suggests the existence of two distinct molecular categories based on their *m*/*z* ratios: amine derivatives and alkylammonium adducts (Appendix A). These compounds arise from the combination of nitrogen with various alkyl fragments during the formation of secondary ions. The presence of amine derivatives is substantiated by the observation of several positive peaks at specific *m*/*z* values, namely 42.04, 44.05, 46.07, 58.07, 296.33, 324.35, 338.38, 352.39, 366.40, 380.42, 394.32, 408.45, 422.47, 464.50, 504.59, 536.59, and 562.60. These peaks align with molecular cations such as C_2_H_4_N^+^, C_2_H_6_N^+^, C_2_H_8_N^+^, C_3_H_8_N^+^, C_20_H_42_N^+^, C_22_H_46_N^+^, C_23_H_48_N^+^, C_24_H_50_N^+^, C_25_H_52_N^+^, C_26_H_54_N^+^, C_27_H_56_N^+^, C_28_H_58_N^+^, C_29_H_60_N^+^, C_32_H_66_N^+^, C_35_H_70_N^+^, C_37_H_76_N^+^, and C_39_H_80_N^+^, respectively. These observations are consistent with the formation of [M + N]^+^ adducts, where M represents a linear or branched hydrocarbon [54]. The presence of hydrocarbons identified through GC-MS analysis, such as C_20_H_42_, C_22_H_46_, C_23_H_48_, C_24_H_50_, C_25_H_52_, C_26_H_54_, C_27_H_56_, C_28_H_58_, C_29_H_60_, C_32_H_66_, C_35_H_70_, C_37_H_76_, and C_39_H_80_N, further supports this relationship.

In contrast, *m*/*z* peaks at 170.20, 284.33, 368.42, 388.39, 438.49, 494.56, 522.59, and 550.62 have been attributed to C_11_H_24_N^+^, C_19_H_42_N^+^, C_25_H_54_N^+^, C_27_H_50_N^+^, C_30_H_64_N^+^, C_34_H_72_N^+^, C_36_H_76_N^+^, and C_38_H_80_N^+^, respectively (Appendix A). These peaks correspond to a series of quaternary alkyl-bearing ammonium adducts frequently encountered in the ToF-SIMS analysis of nitrogen-enriched sedimentary materials [34]. The formation of these molecular cations is likely the result of hydrocarbon chain combinations with dimethylammonium (C_2_H_8_N^+^) and, to a lesser extent, ammonium (NH_4_^+^), both of which are identified as major ions in the TA1, TA2, and TA3 spectra. Furthermore, various *m*/*z* peaks observed at 104.11, 430.38, 640.60, 666.61, 668.60, and 696.65 correspond to organic cations containing nitrogen and oxygen, such as C_5_H_14_NO^+^, C_25_H_52_NO_4_^+^, C_39_H_78_NO_5_^+^, C_41_H_80_NO_5_^+^, C_41_H_82_NO_5_^+^, and C_43_H_86_NO_5_^+^, respectively (Figure 12A). However, the observed fragment distribution also aligns well with the fragmentation patterns of certain sphingolipids, such as ceramides containing 18:1(8E)-sphinganine and 23:0-ceramide [60].

## 4. Discussion

The use of complementary spectroscopic techniques, namely Gas Chromatography–Mass Spectrometry (GC-MS) and Time-of-Flight Secondary Ion Mass Spectrometry (ToF-SIMS), yields a more comprehensive understanding of the organic composition of the Cambrian samples. The application of GC-MS on sample 95E5/2 enables the quantitative molecular analysis of various fatty acids, alcohols, and sterols. This technique not only facilitates compound quantification, but also provides insights into compound sources and their degree of maturation [61]. Further, when combined with the isotopic composition at the molecular level (i.e., CSIA), additional information about the metabolic pathways involved in the biosynthesis of the lipidic compounds can be achieved. In contrast, ToF-SIMS offers valuable insights into the distribution of additional biomolecules associated with lipid compounds in sample 95E5/3, such as glycerides, amino acids, and peptidic fragments [53,56], which are typically not detected by GC-MS. Furthermore, the spatial information obtained through ToF-SIMS contributes to the understanding of the syngenicity versus allogeneity of the organic compounds concerning the rock texture and microstructure [34].

By separately and comparatively interpreting the GC-MS and ToF-SIMS results, a complementary and comprehensive synthesis of the findings can be achieved. Consequently, the discussion section of this study will be divided into three parts. The first two parts will focus on the interpretation of the GC-MS and ToF-SIMS results, respectively, highlighting their respective implications and insights. Subsequently, the third part will compare, integrate, and combine the molecular results from both analyses, facilitating a more comprehensive understanding of the organic composition, biological sources, and their significance in the context of the Cambrian samples.

### 4.1. GC-MS Data Interpretation

The distribution of *n*-fatty acids (*n*-FAs) with ACL (18) and LMW/HMW (5.5) ratios is consistent with a contribution of prokaryotic over eukaryotic sources to the 95E5/2 sample, possibly explained by a recent or modern active soil microbiota [46]. In this regard, the clear prevalence of even over odd-numbered carbons (i.e., CPI of 3.7) among the *n*-FAs series, where C_16_, C_18_, and C_14_ are the most abundant congeners, suggests a dominant bacterial source of the *n*-FAs from both autotrophic and chemoheterotrophic organisms [45,46,62], which are involved in the biogeochemical recycling in soil [63,64,65]. However, a minor presence of HMW *n*-FAs, mostly C_24_ and C_26_, would also mean some input from terrestrial sources (i.e., plants) [66,67,68]. Furthermore, the distribution of *n*-alkanols does not exhibit a clear predominance of either HMW or LMW compounds, indicating a mixture of biological sources [69]. Additionally, the prevalence of even-over-odd-numbered compounds in the *n*-alkanols series denotes a high thermal immaturity and, consequently, suggests recent or modern microbial (C_16_ to C_22_) [70,71] and plant (C_22_ to C_30_) inputs [46,72,73].

The coexistence of both odd- and even-numbered fatty acids implies the presence of multiple microbial groups [74]. The diverse array of iso- and anteiso-methyl-branched compounds, encompassing chain lengths from C_12_ to C_26_, suggests the involvement of various heterotrophic bacteria [62,75] and possibly some fungi in the sample [76]. The prominence of C_15_ and C_17_ iso/anteiso fatty acids specifically indicates potential sources from Gram-positive bacteria which are very common in soils [62,77]. Other distinctive biomarkers provide insights into additional microbial diversity, for instance, the presence of C_18:1*ω*9_ fatty acid indicate fungi or Gram-positive bacteria origin [78], and 10-methyl undecanoic acid (10Me-C_11_) and methyl 10-methyl-hexadecanoate (10Me-C_16_) suggests the involvement of Actinobacteria [79,80].

The sterols, generally sourced from Eukaryotes [81], were extracted from the Cambrian host rock. They most plausibly derive from contemporary soil microbial communities and vegetation native to the sampling area. Cholesterol and ergosterol derivatives indicate a fungal presence, including ubiquitous ectomycorrhizal and saprotrophic fungi in soil conditions [82,83]. The plant sterols campesterol, stigmastenol and stigmastanol, likewise, suggest an input from current surface vegetation through penetration of roots that laterally support and spread fungi and bacterial [84] colonies underground. Intriguingly, cholestadienol has both fungal and plant biosynthetic routes [85], but also different organisms including protozoa, invertebrates or algae [86]. Overall, these sterols derive from lively contemporary soil and endolithic microbiota. Consequently, the presence in sample 95E5/2 of β-sterols, *n*-FAs and *n*-alkanols with high CPI values and even-over-odd predominance together suggests an organic supply attributed to recent or modern microbial activity, supplied by terrestrial communities dominated by plants.

In principle, the wider range of δ^13^C recorded for the *n*-FAs (−26.5 to −36.6‰; Figure 5), suggested the contribution of two different sources, one of prokaryotic sources (i.e., LMW *n*-FAs of δ^13^C values from −29.0 to −26.1‰), and another of eukaryotic phototrophs (e.g., higher plants and microalgae) producing HMW *n*-FAs of relatively more depleted δ^13^C values (−30.6 to −36.1‰) by the use of the C3 Calvin pathway [87]. The *n*-fatty acid carbon isotope values obtained from the linear fatty acids span a relatively wide range from C_14_ to C_26_ (approximately −27‰ to −36‰). In general, depleted δ^13^C values in long-chain fatty acids are indicators of a vegetative origin, while less negative values suggest microbial reworking [88]. Thus, the fatty acid CSIA profile suggests these taxa are largely heterotrophic, assimilating breakdown products of C3 plant materials for growth substrates [89]. Meanwhile, the lighter C_21_ to C_26_ δ^13^C values implicate plant wax cuticular contribution [90], likely from overlying C3 herbaceous vegetation inputs into the soil habitat housing these bacteria and fungi. Overall, the δ^13^C diversity denotes microbial derivation of *n*-fatty acids from taxonomically mixed C3 crop residues and plants surrounding the immediate environment [91,92]. Analyses of intact polar lipid isotope ratios could confirm relative autotrophic activity in this detritus-based food web. The isotopic composition of ^13^C in *n*-FAs provides compelling evidence of the microbial community’s dependence on exogenous inputs originating from terrestrial photosynthetic plants and microbes. These sources are presumed to serve as primary contributors to both soil and endolithic communities.

### 4.2. ToF-SIMS Data Interpretation

The spatial distribution of the organic compounds, as inferred by ToF-SIMS analysis, reveals distinct origins for different molecular groups. The presence of various organic compounds within molecular groups G1 to G4 (Figure 9) suggests an association of lipids from the endolithic microbial community activity. The carbonatic texture observed corresponds to a micrite matrix embedding phosphatic particles (Figure 6A). The biomolecules produced by the endolithic community occur at the phosphatic particle and carbonatic matrix boundary, where it might have a microsite to settle and grow. Among the lipids, glycerolipids and steroids exhibit higher intensities in the molecular group mappings associated with the carbonatic matrix, but contacting with the phosphatic element (Figure 9A). This glycerolipid and sterol association implies that they primarily originate from eukaryotic organisms (e.g., fungi), derived from activities of the microbial endolithic community pioneering the soil formation. The existence of lipids within the phosphate and carbonate rock matrices suggests that meteoric water percolates through soil layers and rock fractures, eventually accessing and depositing organic compounds along boundary zones where phosphatic and carbonate sediments adjoin.

The lipid association is greatly dominated by microbial content, as recognized in the distribution of *n*-FAs and glycerolipids towards a microbial profile (Figure 10A–C). The distribution of *n*-FAs characterized by ToF-SIMS aligns comparably to the distribution obtained from the GC-MS analysis (see Figure 3A), further supporting their common origin within the rock matrix. Moreover, the distribution of diglycerolipids, with a peak at DG (32:0), also corresponds well with the *n*-FA distribution (Figure 10A–C), suggesting their shared origin. The most abundant diglycerides are identified as *n*-C16 hexadecanoate esters. Additionally, the presence of DG (30:0) to DG (36:0) is supported by the occurrence of FAs ranging from *n*-C_14_ to *n*-C_18_, indicating a significant input from bacteria and fungi [34], which are common organisms in soils.

Furthermore, the distribution of *n*-FAs, as observed in ToF-SIMS analysis, also indicates a minor occurrence of HWM *n*-FAs, consistent with the distribution of *n*-FAs obtained by GC-MS analysis (Figure 3A and Figure 10B). The presence of HWM *n*-FAs and their corresponding glycerides, such as DG (40:0), could be attributed to the input of surface vegetation [93]. Additionally, the minor contribution of HWM *n*-FAs may also originate from organic compounds formed in soil and subsequently transported to the rock matrix [94].

The ToF-SIMS analysis has also revealed the presence of various fragment sizes that correspond well with hopanoids and steroids (Appendix A). Even the low intensity of hopanoid fragments hindering their mapping on the sample surface, they could still provide information about possible sources, which are aerobic and anaerobia bacteria [95,96,97]. However, the positive ions associated with steroids exhibit a higher concentration, enabling the characterization of their distribution within the sample. The distribution pattern of the glycerides aligns with their predominant location within the micritic matrix (Figure 9A, Figure 12 and Figure 13). This supports the notion that the glycerides originate from a shared biological source, specifically the fungal component of the endolithic community. However, there is also likely some additional supply coming from the vegetal community in the surrounding soil. The fragmentation pattern of the detected steroids is consistent with the presence of cholestadienol, cholesterol, ergostadienol, ergosterol, stigmasterol and stigmastanol. These compounds originate from diverse eukaryotic organisms including fungi, plants and algae [84,98,99,100,101,102]. As such, the steroid profile demonstrates the presence of these specific compounds sourced from the complex endolithic microbial community.

Interestingly, a distinct group of lipids, known as wax esters [58,59], exhibit a unique distribution pattern within the sedimentary rock. These wax esters are found in specific microsites, present in both phosphatic and micritic matrices (see Figure 9C). Wax esters can originate from a variety of organisms [58,103,104], including kinds of bacteria [105,106,107,108]. Their localization and particular composition in the sample 95E5/3, characterized by short-chained structures, imply that they were formed through the activities of specific microbial populations that occupied particular niches within the phosphatic and carbonatic matrices. It is likely that these wax esters were accumulated in the sediment as a result of chemosynthetic respiration or heterotrophic metabolism processes facilitated by these microbial communities [104].

In contrast, another group of N-bearing compounds (Figure 9D) display a significantly broader distribution throughout the analyzed sample, indicating that they originate from diverse sources. The widespread occurrence of these compounds within the phosphatic and carbonatic matrices suggests their association with more recalcitrant materials that have undergone degradation and subsequent accumulation within the soil [109]. The presence of nitrogen-bearing compounds, specifically trialkylamines organized in filamentous structures, implies a potential association with fungal hyphae [34]. Furthermore, the fractionation pattern observed in some N-bearing compound fragments aligns with that of ceramides, which are highly prevalent compounds in fungi [34,60]. The nitrogen-bearing ions and adducts observed in the rock interior are thought primarily derived from ammonification processes, which involve the microbial-driven breakdown of proteins and amino acids in soil [109]. During these processes, proteins and amino acids undergo hydrolysis and decarboxylation, resulting in the release of ammonium ions and various nitrogen-containing compounds [110]. The presence of nitrogen-bearing compounds in the rock interior not only reflects the input of nitrogen from biological sources, but also indicates the active biogeochemical processes occurring within the soil.

### 4.3. Data Synthesis: Organic Matter Origin

The dominant molecular profile observed in the Cambrian samples bears multiple hallmarks of fresh organic matter inputs from active modern soil microbial communities and vegetation. The prevalence of diverse bacterial and fungal lipids in both samples reveals recent community activity. The prominence of C_16_, C_18_, and C_14_ chain lengths FAs identified in both samples fits common bacterial lipids [62]. Notably, iso and anteiso methyl branched fatty acids, which are characteristic of heterotrophic bacteria thriving in soils [62], were identified in 95E5/2, revealing recent community activity. Additionally, the associated distribution of glycerolipids like diglycerides, alongside fatty acids identified in the sample 95E5/3, points to the breakdown of membrane lipids sourced from the decay of cells. The occurrence of abundant fungal ergosterols in sample 95E5/3 and cholesterol in both samples likewise demonstrates the incorporation of modern eukaryotic biomass [82]. Plant-derived compounds such as campesterol and stigmastanol in sample 95E5/2 as well as stigmasterol in both samples further showcase inputs from contemporary vegetation likely populating the surrounding region [85,111,112,113]. Thermal immaturity indicators across the multiple biomarkers in sample 95E5/2 indicate relatively recent biological production, rather than ancient diagenesis. Moreover, carbon isotope fractionations of sample 95E5/2 verify dependence on C3 plants and microbes prevalent in modern soils [92]. Together, these substantive indicators negate sole derivation from preserved Cambrian organic matter, strongly supporting pronounced molecular contributions from thriving soil communities that presently infiltrate and subsist within fractures in the exposed rock. Detailed comparison between organics and their sources (some could be defined to microbial taxa level) in the two studied samples were described in Appendix A.

Comparison of the molecular profile from the Cambrian rock to lipid assemblages characterized in other ancient deposits in southern Spain provides further evidence that much of this organic material reflects recent microbiota rather than preservation over geological timescales. Prior studies of paleosol formations dated to the early Pleistocene [32,114,115], which should exhibit distinct signs of thermal maturation among hydrocarbon biomarkers [116]. However, the rock samples here contain an abundance of microbiota-derived lipids in sample 95E5/2 retain strong even-over-odd predominance. The prominent lack of diagenetic alterations across compound classes implies origins from extant soil inhabitants rather than ancient preservation regimes. Intriguingly, while the rock lipids denote a recent or modern microbial consortium, overt physical indicators of extant inhabitants remain largely unobserved. The glycerolipids, fatty acids, and sterols appear finely dispersed within the phosphate and carbonate mineral matrix (sample 95E5/3, Figure 9A–C). Such distributions could arise through transport and accumulation along fractures in addition to in situ release during endolithic boring. Prior mineral dissolution enabling cavity formation seemingly occurred proximal to the present day, as matrices show limited subsequent re-precipitation. Nonetheless, filamentous morphologies among certain nitrogen-bearing compounds (sample 95E5/3, Figure 9D) present microscopic evidence that is directly consistent with fungal hyphae dimensions [34]. These putative delicate physical traces raise prospects for better-preserved structural biomarkers lingering amid the Cambrian strata. Elucidating the continuity and extremity of such remnant organic frameworks could help constrain the timing of the most recent inoculation events. Even if some relic cells persist, the investigated molecular profile in both samples retains an unambiguous signature of thriving subsurface microbiota rooted in modern surface processes. Additional high-resolution microbial imaging approaches are imperative to fully reveal the in situ physical constructs reflecting this recent soil microbial activity.

Furthermore, the detection of extensive nitrogen compound distributions in the host rock sample 95E5/3 provides evidence that active nitrogen cycling is occurring in this subsurface habitat [117]. Bacteria drive ammonification and nitrification reactions, producing substrates that can facilitate fungal growth. In turn, the identified fungal lipids (e.g., ceramides and sterols) suggest eukaryotic utilization of this bioavailable nitrogen [118]. The nitrogen cycling in the microbial community intimates opportunities for symbiotic nutrient exchange. Bacterial–fungal symbiosis in soils and endolithic habitats often emerges from nutrient cycling, whereby microbes exchange metabolites and growth factors. Such associations, deriving multifaceted rewards through its fungal co-cultivation centered on accessing otherwise unavailable nitrogen, phosphorus, carbon, and ribose substrates [119]. Attenuation of bacterial starvation responses through nutritional supplementation facilitates the prevalent soil taxon cross-kingdom partnerships observed.

Moreover, the detected microbial-mineral interactions in sample 95E5/3 further add complexity to the endolithic symbiosis. The prominent oxalic acid fragments found in the spectra (e.g., C_2_HO_4_^−^) denote active weathering whereby bacteria liberate growth-limiting nutrients from the crystalline host to fuel biogeochemical cycling [13,119]. Such lithotrophic augmentations of substrate bioavailability may promote conditions enabling stable coexistence [120,121]. The soil microbes enable the later establishment of plant communities. Plants have a vital role in soil formation and development, since plant root exudates and decomposing litter make up the principal sources of organic matter [122], which can be utilized by microbes in return. This intricate trophic pathway, originating from the breakdown of bedrock to the propagation of symbionts, unveils interconnected mechanisms throughout various stages of community assembly [123]. Ultimately, plants and their root-associated microbes form positive feedback, leading to closed nutrient cycles and accelerated soil formation and development.

## 5. Conclusions

The molecular analyses provided significant insights into the soil microbial communities within the Cambrian host rock. The prevalence of bacteria and fungi is evidenced through distributions of fatty acids, sterols, wax esters, and nitrogenous compounds. Carbon isotopic signatures indicate reliance on C3 vegetation, while spatial mappings reveal niche chemosynthetic processes. The molecular profile observed in the Cambrian subsurface rock delivers key glimpses into the initial stages of soil genesis and development in the late Cenozoic and Quaternary sedimentary record. The diversity of lipids implies complex microbial communities, which likely originated from overlying vegetative litter, and plant roots penetrated fractures in the rock. Their metabolic activities encompass heterotrophy, chemosynthesis, and ammonification, reflecting pivotal initial steps of microorganisms in cooperating with plants to form and develop soils through alteration of ancient rock. Furthermore, this study provides molecular clues substantiating nutrient-centered symbioses within endolithic microbial communities, where nitrogen and phosphorous may play an essential role. The documented metabolic interdependence potentially enabling ecological niches suggests such cooperative substrate cycling may commonly promote survival in nutrient-limited subsurface habitats. As such, these molecular fingerprints capture the pioneer consortia and processes underpinning the pedogenic transformation of shallow subsurface sediments into nascent soils and microbial habitats during recent Earth history. The relative abundances of compounds can denote primary colonizers, while their spatial mapping traces early horizonation patterns.

Critically, these profiles showcase how contemporary microbiota can actively penetrate and inhabit ancient subsurface rock environments, thus the results provide an important cautionary tale regarding the potential for organic contamination of geological samples by modern or recent microbial communities. Such contamination, if not properly identified, can impede or obscure interpretation of ancient biomarkers hosted in Proterozoic or Paleozoic rocks, posing challenges for paleoenvironmental reconstruction relying on molecular biosignatures, as well as analysis of extraterrestrial samples. Proper containment procedures and molecular screening will be essential to minimize the contamination of extraterrestrial return samples. Moving forward, coupling mineralogical context with organic characterization during paleoenvironmental examinations can help disentangle modern soil microbial inputs from ancient molecular signals. These best practices will significantly advance efforts to elucidate early Earth or potential extraterrestrial biosignatures, while accounting for ubiquitous intervention of soil microorganisms.

## Figures and Tables

**Figure 1 microorganisms-12-00513-f001:**
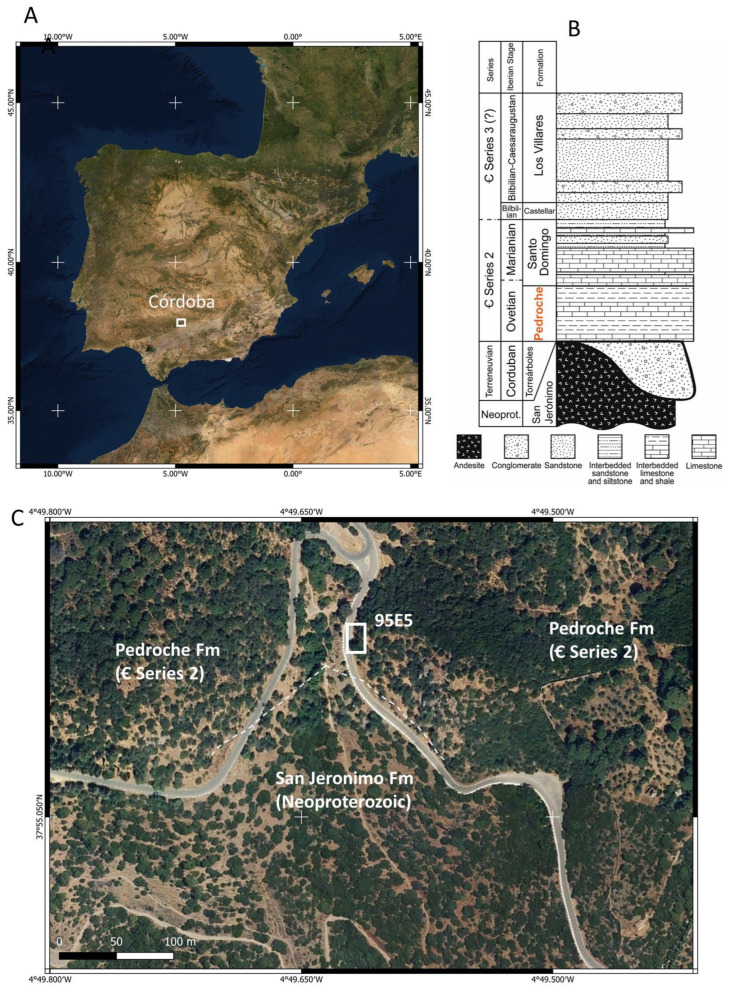
General location and settings of the Lower Cambrian Pedroche Formation outcrops. (**A**) The Iberian Peninsula, with emphasis on the study area. (**B**) Chronostratigraphic section of the Sierra de Córdoba region, illustrating the major stratigraphical units, including the Lower Cambrian Pedroche Formation examined in this study. (**C**) Image of the Pedroche Formation in the Las Ermitas outcrops contacting the San Jeronimo Formation (terminal Precambrian to early Cambrian) through a major discordance. Section 95E5, where samples were collected, is traced as a white square. The satellite image shows the presence of a Mediterranean forest, which density decreases and clears in the stepper areas of the topographic relief.

**Figure 2 microorganisms-12-00513-f002:**
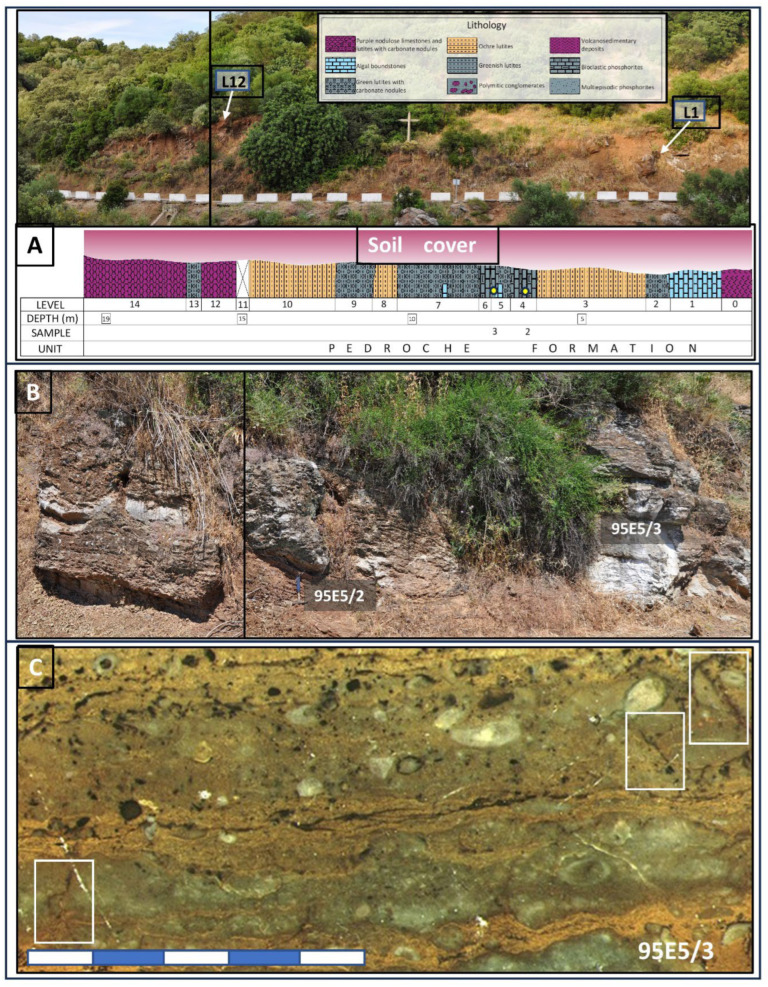
Showcases of the Pedroche Fm outcrops in section 95E5, which served as the sampling location. (**A**) The Cambrian geological formations are overlain by a soil layer approximately 50 to 75 cm thick, a consequence of road construction leading to the Las Ermitas sanctuary. Reference points for Levels 1 and 12 (L1 and L12) are demarcated to provide context within the lithological section. Specific samples, 95E5/2 and 95E5/3 are denoted by small yellow dots in the corresponding stratigraphic section below. In (**B**), a focused depiction of the carbonatic levels sampled for 95E5/2 and 95E5/3 is presented. Furthermore, (**C**) showcases a polished section revealing sample 95E5/3, wherein centimeter- to millimeter-sized cracks, delineated by white squares, enhance fluid flow within the rock matrix.

**Figure 3 microorganisms-12-00513-f003:**
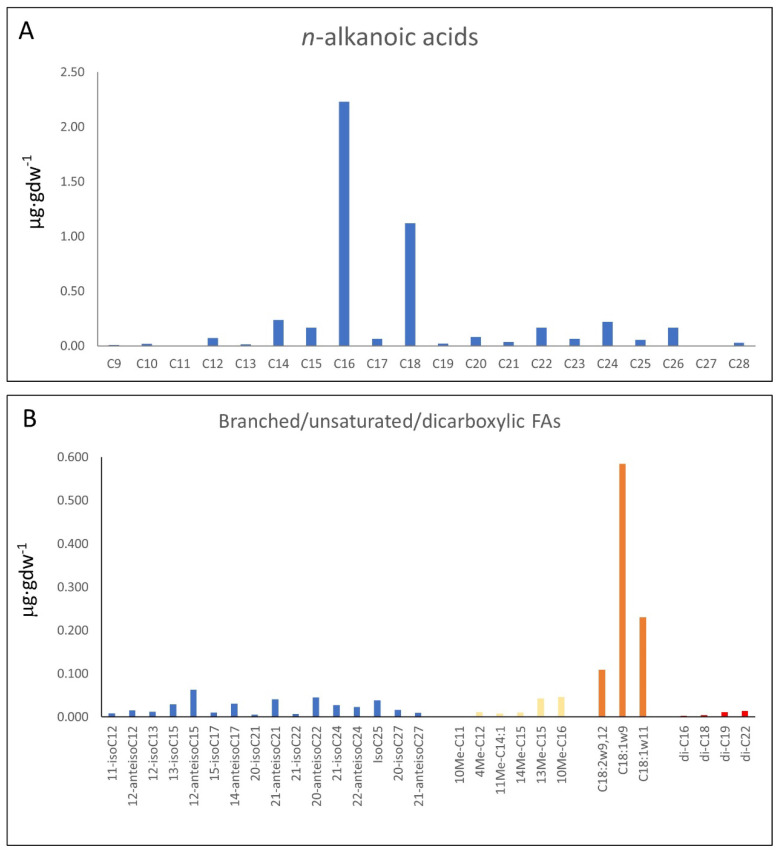
Molecular distribution of *n*-alkanoic or fatty acids (*n*-FAs) (**A**) and branched/ unsaturated/dicarboxylic fatty acids (**B**) analyzed by CG-MS.

**Figure 4 microorganisms-12-00513-f004:**
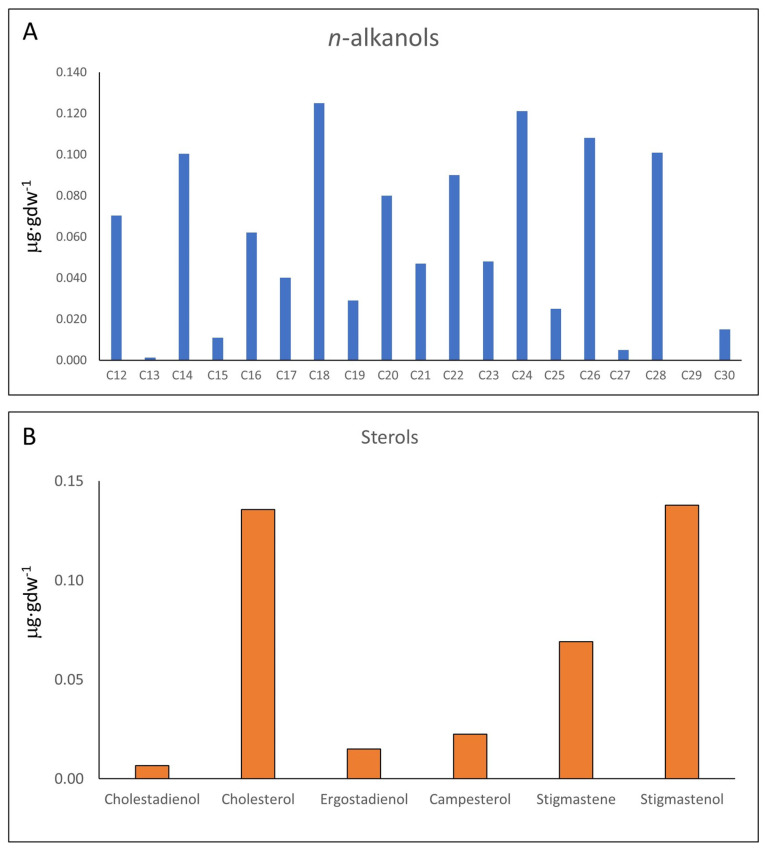
Supplementary GC-MS findings depicting (**A**) concentrations of *n*-alkanols, which display a more uniform profile, indicative of heterogenous sources comprising prokaryotic and eukaryotic organisms, including multicellular organisms such as marine invertebrates, algae, and terrestrial plants; and (**B**) sterol concentrations, including cholestadienol (cholesta-4,6-dien-3-ol, (3β)-), cholesterol, ergostadienol (ergosta-5,22-dien-3-ol, acetate, (3β,22E)-), campesterol, stigmastene (stigmast-5-ene, 3β-(trimethylsiloxy)-, (24S)-), and stigmastanol.

**Figure 5 microorganisms-12-00513-f005:**
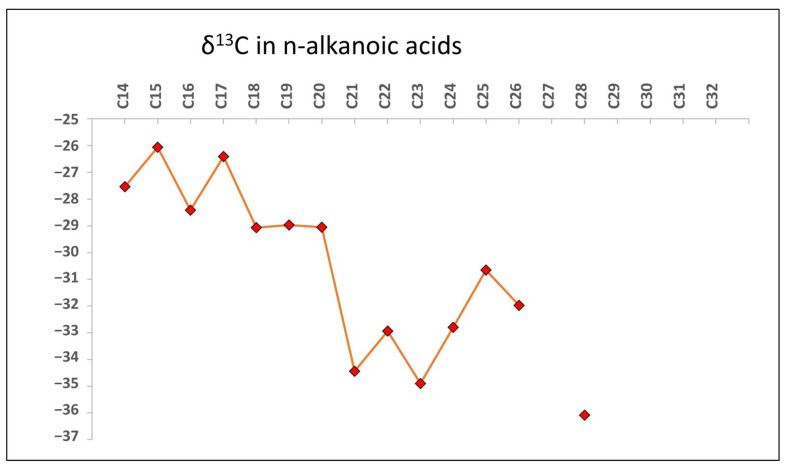
Scatter plot illustrating the variable δ^13^C values (average δ^13^C = −31‰) from extracted soil fatty acid esters ranging from C_14_ to C_28_ chain lengths implicate origins from C3 terrestrial vegetation litter undergoing remineralization by active aerobic heterotrophs. ^13^C-depleted wax ester isotopes agree with woody and non-woody plant residue contributions rather than aquatic sources. Selective chain length dominance by even carbon numbers further points to cuticular plant wax supplies mobilized into rock fractures and fissures by percolating fluids.

**Figure 6 microorganisms-12-00513-f006:**
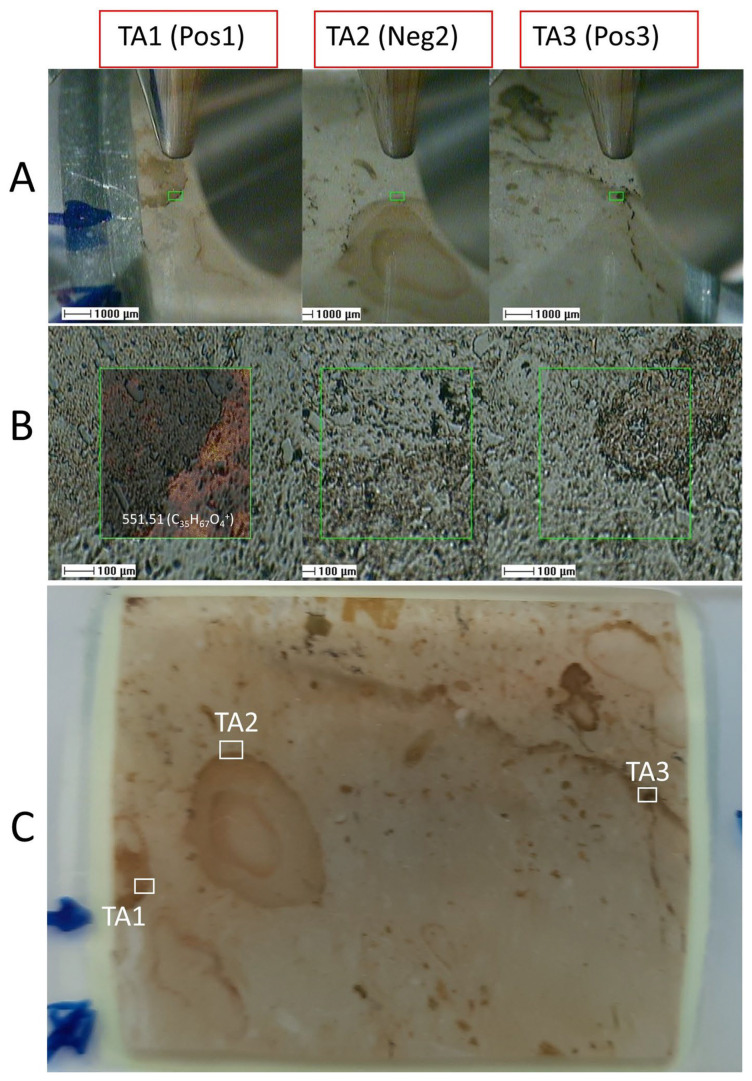
Spatial positioning of the three distinct target areas (TAs) within the sample 95E5/3 for organic analysis using the Time-of-Flight Secondary Ion Mass Spectrometry (ToF-SIMS) technique. TAs include TA1 (positive ions), TA2 (negative ions), and TA3 (positive ions). (**A**) Precise targeting of TA1 to TA3 regions facilitated by the integration of a visible light camera with the ToF-SIMS apparatus. (**B**) Delimitation of 500-micron squared TAs accomplished through the use of a scanning electron microscope (SEM) probe in the ToF-SIMS setup, with an example depicting the mapping of *m*/*z* 551.54 corresponding to a lipid fragment over the TA1 SEM image. (**C**) Identification and visualization of the specific TAs (TA1, TA2, and TA3) selected for comprehensive organic analysis via ToF-SIMS, encompassing phosphatic and carbonatic matrices, enabling the assessment of variations in both organic and inorganic composition within the sample.

**Figure 7 microorganisms-12-00513-f007:**
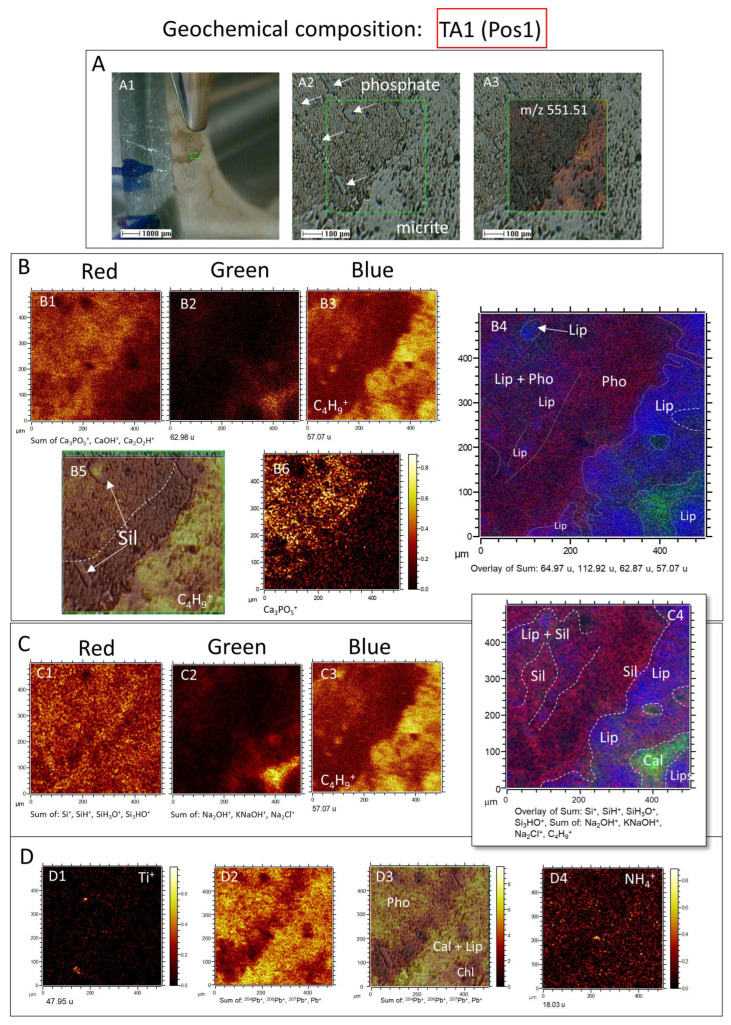
Geochemical composition of the TA1 area in sample 95E5/3 analyzed using the positive ion mode of ToF-SIMS. (**A**) Detailed characterization of TA1, including (**A1**) visualization using the visible camera and ToF-SIMS SEM probe, (**A2**) identification of textural variations between the phosphatic and carbonatic matrices (with white arrows highlighting traces of a silicified archaeocyathid wall), and (**A3**) molecular mapping of *m*/*z* 551.51. (**B**) Red–Green–Blue (RGB) overlay displaying (**B1**) PO_3,_ and Ca cationic complexes associated with the phosphatic matrix, (**B2**) Na ion complexes likely representing mineralized silica or phyllosilicate, and (**B3**) C_4_H_9_^+^ ion resulting from lipid fragmentation. The resulting overlay image (**B4**) depicts the spatial distribution of these three distinct fragments within the sample, with (**B5**) and (**B6**) indicating the occurrence of lipids in relation to the phosphatic mineral matrix. (**C**) RGB overlay demonstrating (**C1**) silica represented by the sum of Si^+^, SiH_3_O^+^, and Si_3_HO^+^ fragments, (**C2**) the sum of Na, K, and Cl cation complexes, and (**C3**) lipids represented by the C_4_H_9_^+^ fragment, with the overlay image (**C4**) illustrating the distribution of lipids in relation to silica and phyllosilicates. (**D**) Distribution mapping of (**D1**) Ti^+^ ions, (**D2**) various isotopes of Pb, (**D3**) cropping of Pb isotope mapping over the ToF-SIMS SEM image, and (**D4**) occurrence of ammonium indicating the presence of N-bearing compounds such as amines.

**Figure 8 microorganisms-12-00513-f008:**
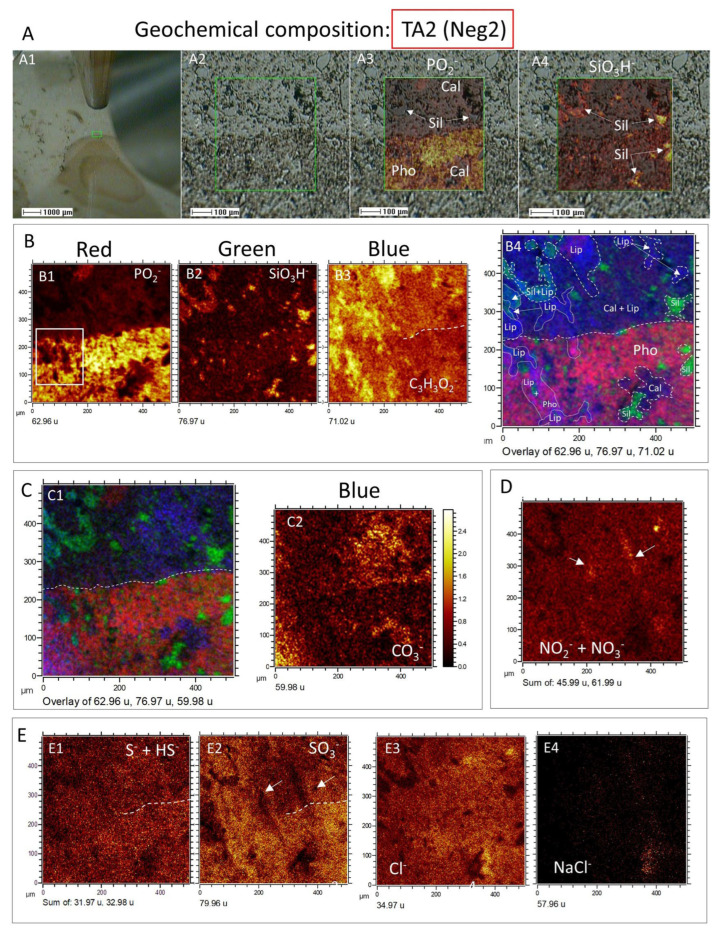
Geochemical composition of the TA2 area in sample 95E5/3 analyzed using the negative ion mode of ToF-SIMS. (**A**) Detailed characterization of TA2, encompassing (**A1**) visualization using the visible camera and ToF-SIMS SEM image, (**A2**) identification of textural variations between the phosphatic and carbonatic matrices, (**A3**) mapping of PO_2_^−^ over the SEM texture image to trace phosphate distribution, and (**A4**) mapping of SiO_3_H^−^ over the SEM texture image to depict silica distribution. (**B**) Red–Green–Blue (RGB) overlay exhibiting (**B1**) PO_2_^−^, (**B2**) SiO_3_H^−^, and (**B3**) C_3_H_3_O_2_^−^, resulting in (**B4**) a composite map illustrating the distribution of phosphate, silica, and lipids within TA2. (**C**) Overlay image (**C1**) demonstrating the interrelationship among phosphate, silica, and carbonate in TA2 by combining (**B1**) PO_2_^−^, (**B2**) SiO_3_H^−^, and (**C2**) CO_3_^−^ corresponding to the distribution of the carbonatic matrix. (**D**) Sum of NO_2_^−^ and NO_3_^−^, indicating linear structures unrelated to the TA2 texture. (**E**) Mapping of (**E1**) S^−^ and HS^−^, (**E2**) SO_3_^−^, (**E3**) Cl^−^, and (**E4**) NaCl^−^. The arrows in (**D**) indicate the concentrated distribution of NO_2_^−^ and NO_3_^−^ ions, while in the SO_3_^−^ distribution map (**E2**), the areas enhanced in NO_2_^−^ and NO_3_^−^ appointed by the arrows, show vacancies.

**Figure 9 microorganisms-12-00513-f009:**
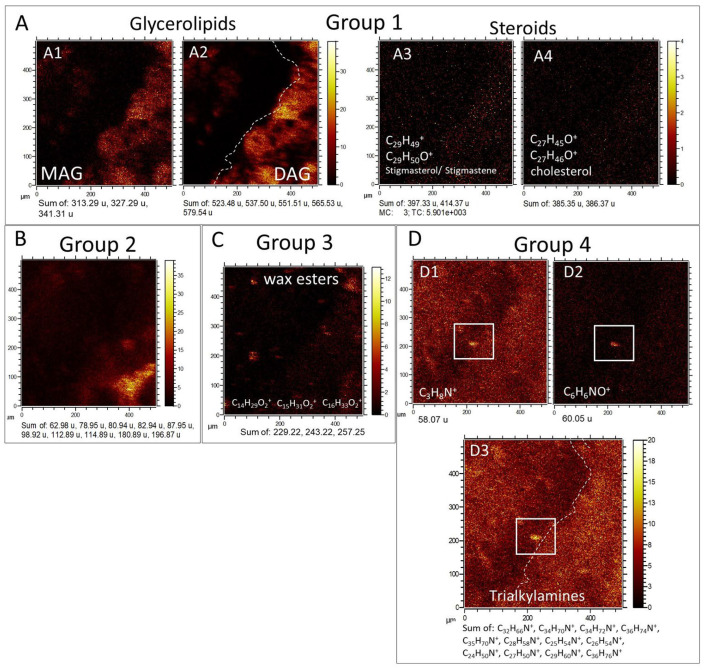
Morphological groups 1 to 4 (G1 to G4) derived from the molecular mapping of positive fragments within the TA1 area of the sedimentary rock sample. (**A**) G1 encompasses a diverse range of lipids, including monoacylglycerides (MAG) (**A1**), diacylglycerides (DAG) (**A2**), and steroids (**A3**,**A4**). These lipid compounds are predominantly observed within the micritic matrix, suggesting a syngenetic origin (refer to Appendix A). (**B**) G2 corresponds to the spatial distribution of inorganic cations resulting from the combination of Na, Mg, K, and Ca with Cl. Interestingly, the lipidic fraction (G1) shows a relative depletion of these inorganic cations. (**C**) G3 represents the distribution pattern of distinct wax esters, such as C_14_H_29_O_2_^+^, C_15_H_31_O_2_^+^, and C_16_H_33_O_2_^+^, suggesting a different origin compared to the majority of lipids. (**D**) G4 is determined by the presence of positive fragments derived from nitrogen-bearing compounds, including the [M − H]^+^ fragment of a propylamine (**D1**), C_6_H_6_NO^+^ likely originating from an aromatic precursor (**D2**), and the cumulative intensity of various trialkylamines (**D3**).

**Figure 10 microorganisms-12-00513-f010:**
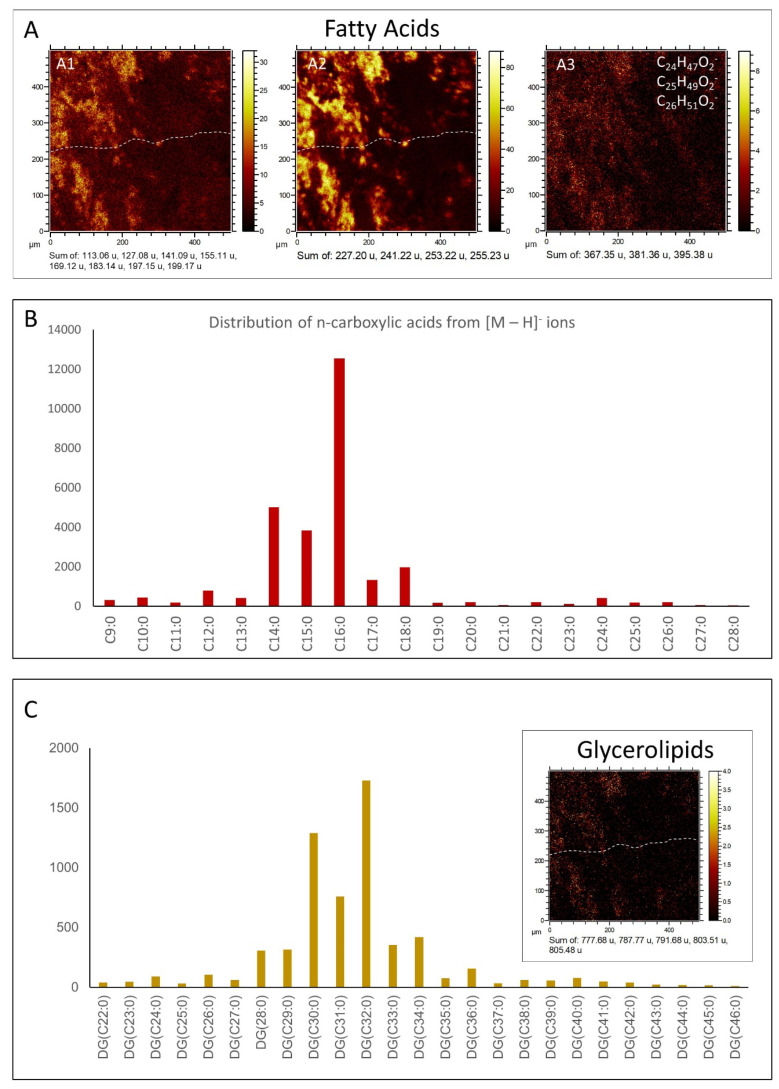
Comparison of the occurrence and spatial distribution of *n*-Fatty Acids (*n*-FAs) in TA2 with the distribution of Diacylglycerides (DAGs) in TA1 using Time-of-Flight Secondary Ion Mass Spectrometry (ToF-SIMS). (**A**) Identification of distinct *n*-FAs, including (**A1**) C6:0 (*m*/*z* 113.06), C7:1 (*m*/*z* 127.08), C8:1 (*m*/*z* 141.09), C9:1 (*m*/*z* 155.11), C10:1 (*m*/*z* 169.12), C11:1 (*m*/*z* 183.14), C12:1 (*m*/*z* 197.15), and C12:0 (*m*/*z* 199.17); (**A2**) C14:0 (*m*/*z* 227.20), C15:0 (*m*/*z* 241.22), C16:1 (*m*/*z* 253.22), and C16:0 (*m*/*z* 255.23); and (**A3**) C24:0 (*m*/*z* 367.35), C25:0 (*m*/*z* 381.36), and C26:0 (*m*/*z* 395.38). (**B**) Analysis of the distribution of *n*-FAs ranging from C9 to C28 utilizing the [M − H]^−^ ions in TA2. (**C**) Distribution pattern of Diacylglycerides (DAGs) based on the presence of [M + H − H_2_O]^+^ ions in TA1, showing a very similar pattern in the distribution of *n*-FAs in TA2.

**Figure 11 microorganisms-12-00513-f011:**
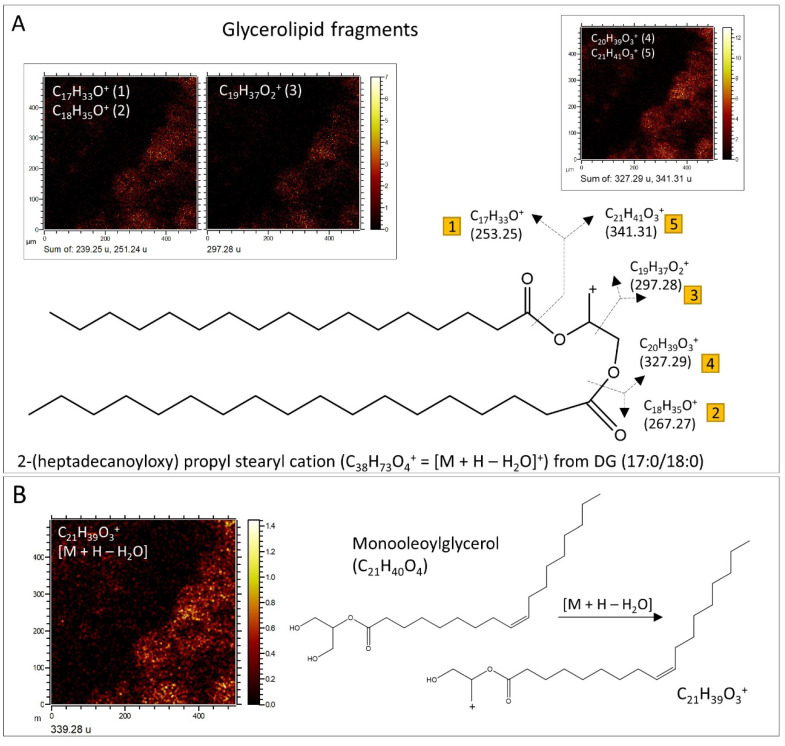
Structural characterization of lipids through occurrence and fractionation patterns. (**A**) Reconstruction of DG (17:0/18:0) based on the occurrence of various fragments within the micritic matrix, following the glycerolipid fragment assignment reference provided in Appendix A. (**B**) Identification of monooleoylglycerol through the presence of its [M + H − H_2_O]^+^ cation within the boundary between the phosphatic element and the carbonatic mineral matrix.

**Figure 12 microorganisms-12-00513-f012:**
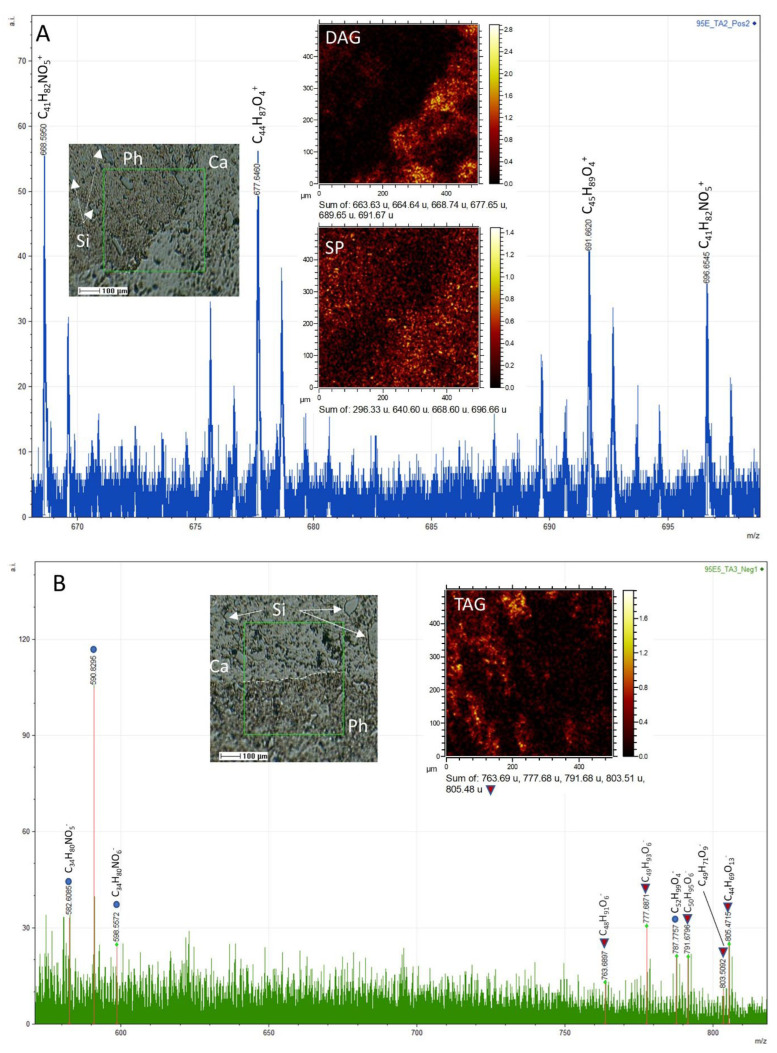
Spectra acquired from compound fragmentation in TA1 and TA2, representing both positive and negative fragments, are presented. (**A**) The mass spectrum of TA1 exhibits prominent *m*/*z* peaks at 668.60, 677.64, 689.65, and 691.69, corresponding to various cations of ceramides. SEM and ToF-SIMS mapping of these ions are included as references, highlighting their similar distribution patterns to lipids. (**B**) Similarly, TA2 displays significant peaks of large negative fragments of glycerides and other lipids at *m*/*z* 763.69, 777.68, 787.77, 791.68, 803.51, and 805.48, assigned to C_48_H_91_O_6_^−^, C_49_H_93_O_6_^−^, C_52_H_94_O_4_^−^, C_50_H_95_O_6_^−^, C_49_H_71_O_9_^−^. Notably, these fragments exhibit the same distribution pattern as fatty acids (refer to Figure 10). In both figures (**A**,**B**), the SEM image is annotated with the letters Si, Ph, and Ca, indicating the presence of silica, phosphate, and calcite, respectively.

**Figure 13 microorganisms-12-00513-f013:**
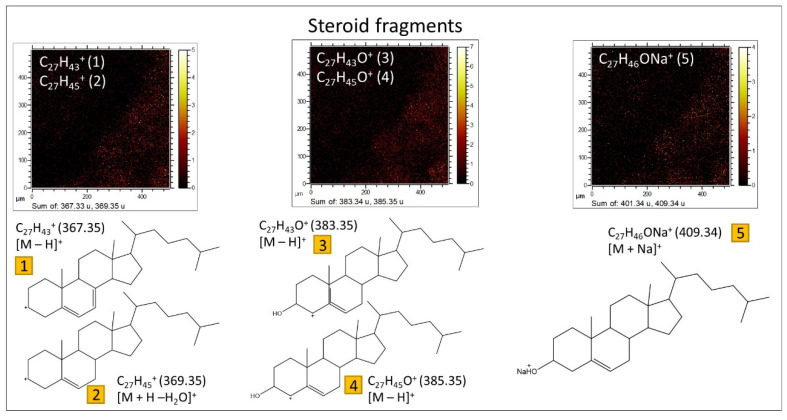
Steroid identification accomplished by detecting specific fragments within the micritic matrix, as indicated in Appendix A.

## Data Availability

Data are contained within the article and Appendix A.

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
