# Peer review of "The Molecular Profile of Soil Microbial Communities Inhabiting a Cambrian Host Rock"

_microorganisms, 2024, doi:10.3390/microorganisms12030513_

Round 1
Reviewer 1 Report
Comments and Suggestions for Authors
The manuscript entitled “Molecular profile of soil microbial communities inhabiting a Cambrian hostrock” is aims to soil genesis on Cambrian hostrock. The authors examined a wide range of biosignatures that make it possible to interpret changes in bacterial and fungal profiles during the process of soil formation. Results underscore the ability of contemporary microbiota to actively inhabit rocky substrates, necessitating rigorous contamination controls when evaluating ancient molecular biosignatures or extraterrestrial materials collected. To my mind this manuscript is topical and corresponding to the aims and scopes of the “Microorganism’ journal and it has important practical and fundamental significance.
Here are the comments I found while reading the manuscript
1. The abstract should more specifically describe the results not only be limited to describing what was done in the study.
2. In the introduction, it is worth providing data on the role of microorganisms in the transformation of clay minerals, which are often found in rocks and contain accessible elements and are the basis for the formation of soils
3. The introduction should more clearly describe why the selected object was studied and why it is interesting.
4. The number of samples is not large, why were only two selected?
5. In the interpretation of data and in conclusion, more attention should be paid to the comparison of the two studied samples, in connection with their possible evolution of soil formation. The authors examined only two samples and it is important to compare them in detail.
6. Based on the interpretation of the results, I have doubts that it is corresponding to the aims and scopes of the “Microorganism’ journal. A detailed description of the analysis of organic markers (MALDI-TOF) and their comparison with fungal and bacterial taxa is needed. In addition, it is worth profiling samples for 16S rRNA genes. In this regard, the name of the manuscript is puzzling.
General conclusion. The article has a significant amount of important and new geochemical data that is well interpreted from the point of view of geochemistry and soil formation. However, the small amount of data on microbial communities raises doubts about the correctness of the selected journal.
Author Response
The manuscript entitled “Molecular profile of soil microbial communities inhabiting a Cambrian hostrock” is aims to soil genesis on Cambrian hostrock. The authors examined a wide range of biosignatures that make it possible to interpret changes in bacterial and fungal profiles during the process of soil formation. Results underscore the ability of contemporary microbiota to actively inhabit rocky substrates, necessitating rigorous contamination controls when evaluating ancient molecular biosignatures or extraterrestrial materials collected. To my mind this manuscript is topical and corresponding to the aims and scopes of the “Microorganism” journal and it has important practical and fundamental significance.
We are very grateful to the reviewer for the positive comments. We’d like to clarify that our paper primarily focuses on utilizing advanced mass spectrometry techniques to detect traces of modern soil microbiota and their activities within ancient rock matrices. The limited discussion on microbial changes during soil formation might have resulted from a potential misunderstanding related to our discussion of “soil formation”. To address this, we have revised the original text by correcting the description of soil formation and emphasizing our main objective—to identify modern soil microbiota from biomolecules preserved within ancient rock matrices. More modifications were made accordingly based on the reviewer’s suggestions below.
Here are the comments I found while reading the manuscript.
- The abstract should more specifically describe the results not only be limited to describing what was done in the study.
Thanks for the reviewer’s advice. We have modified the abstract and added more results on the diverse molecular composition, including alkanols, sterols, fatty acids, glycerolipids, wax ester, and nitrogen-bearing compounds, using both GC-MS and ToF-SIMS techniques. The abstract also included discussions on sources of the organics which suggest a contemporary origin linked to microbial activity in a soil-type ecosystem, highlighting the importance of rigorous contamination controls for assessing ancient molecular biosignatures and extraterrestrial materials.
- In the introduction, it is worth providing data on the role of microorganisms in the transformation of clay minerals, which are often found in rocks and contain accessible elements and are the basis for the formation of soils.
Thanks to the reviewer’s suggestion, we have added the corresponding information in Introduction. We included examples of interactions between clay minerals and microorganisms, as well as the influence of those activities to soils formation. Please refer to line 48 to 58.
- The introduction should more clearly describe why the selected object was studied and why it is interesting.
We appreciate the suggestion from the reviewer. The reason we studied these two samples is that they underwent complex geological and climatic processes during the Ediacaran-Early Cambrian periods. At the beginning of the original research, we hoped to analyze ancient biomarkers preserved in the samples through molecular analysis, in order to reconstruct the microbiological and ecological processes at that time. However, during the experiment and analysis, we found there were many modern biomarkers in the ancient rocks, which might come from the interaction between modern or recent soil microbiota and rock matrices. Therefore, it resulted very necessary to conduct a detailed investigation to distinguish modern soil microorganisms from biomolecules identified in ancient rocks. Hope the explanation is reasonable. We have added the reasons in the Introduction section. Please refer to lines 87 to 95.
- The number of samples is not large, why were only two selected?
We appreciate the reviewer’s comment regarding the number of samples analyzed. Our aim of this study was to demonstrate the feasibility and potential of using GC-MS and ToF-SIMS to detect microorganism traces and metabolites in rock matrices. As the first investigation combining these two analytical techniques for this purpose, analyzing two representative samples allowed us to effectively showcase the methods while keeping the scope focused. The two samples selected contained sufficient diversity in terms of biomolecules and their fragments to validate the usefulness of the techniques. The results successfully demonstrate that GC-MS and ToF-SIMS can identify molecules sourced from soil microbes and plants within the rock matrix. These findings establish a foundation for expanding the research to a wider range and number of samples, now that the utility of the combined approach has been established.
- In the interpretation of data and in conclusion, more attention should be paid to the comparison of the two studied samples, in connection with their possible evolution of soil formation. The authors examined only two samples and it is important to compare them in detail.
Thanks to the reviewer’s suggestion. We have now specified and compared the organics we found in each sample with different techniques, as well as their biological sources. They were described in both Interpretation and Conclusion. Data of the two samples were obtained by GC-MS and ToF-SIMS. The GC-MS tested extracted organics of a quantitative advantage, while the ToF-SIMS conducts in situ analysis of an imaging advantage. They are complementary to each other. Only by combining results from two studied samples we could get a better understanding of the biological processes undergoing in the rocks.
As to the possible evolution of soil formation, we have emphasized on the roles played by indicated biological sources (i.e., bacteria, fungi and plants). Please refer to Discussion, especially from line 833 to 857 for related information.
- Based on the interpretation of the results, I have doubts that it is corresponding to the aims and scopes of the “Microorganism” journal. A detailed description of the analysis of organic markers (MALDI-TOF) and their comparison with fungal and bacterial taxa is needed. In addition, it is worth profiling samples for 16S rRNA genes. In this regard, the name of the manuscript is puzzling.
We appreciate the reviewer’s comments and would like to clarify that the technique used for organic and mineral characterization in the manuscript was Time-of-Flight Secondary Ion Mass Spectrometry (ToF-SIMS) instead of MALDI-TOF. ToF-SIMS provides highly sensitive surface analysis, making it uniquely suited to detect trace biomarkers within rock matrices. While MALDI-TOF is typically used to sequence proteins and map biomolecules in tissues. We feel ToF-SIMS was the most appropriate choice given our objectives and scope, and the data successfully demonstrated the utility of combining it with GC-MS.
According to the reviewer’s suggestion on “detailed description of the analysis of organic markers and their comparison with taxa”, the information was added in Interpretation and listed in Table S9.
As to profiling samples for 16S rRNA genes, we agree it is an excellent idea and would like to conduct the work in future’s dedicated research, along with sensitive techniques such as CARD-FISH to identify active species in the ancient rock.
We believe our study aligns well with the aims and scopes of “Microorganisms” journal. This journal covers microbe-mineral interactions and environmental microbiology topics. Our work used advanced mass spectrometry techniques to evaluate whether signatures of modern soil microbes could be detected in ancient subsurface rocks. These molecular fingerprints provide evidence for the penetration and pioneering colonization of deep rocky substrates by surface microbial communities. Therefore, this research advances fundamental knowledge about the ecology, habitat range, and mineral interactions of microorganisms in natural environments. It reveals an intriguing portal connecting surface and subsurface realms through microbial migration and rock colonization. These novel insights align with the journal’s goals of elucidating microbial dynamics in natural and built ecosystems across scales, while engaging microbial specialization and diversification across gradients.
General conclusion. The article has a significant amount of important and new geochemical data that is well interpreted from the point of view of geochemistry and soil formation. However, the small amount of data on microbial communities raises doubts about the correctness of the selected journal.
We appreciate the constructive feedback from the reviewer. As the reviewer suggested, we re-examined the experimental data and results on microbial communities and now explained and discussed more on this topic. Please refer to Interpretation and Conclusion.
Reviewer 2 Report
Comments and Suggestions for Authors
The manuscript brings interesting results regarding the weathering process at microbiological scale. By highlighting the role of microbiota in pedogenesis processus. The introduction is well written. However, the references have to be cited following the journal standards, and the main manuscript need be structured following the section suggested by Authors Guide (Introduction, Materials and Methods, Results, Discussion, and Conclusions). The results are well presented and interpreted.
Find below some specific comments:
Line 85: the author needs to be consistent in using the abbreviation GC-MS.
Line 159: add the coordinate of the sampling points.
Comments on the Quality of English Language
Minor language correction is needed.
Author Response
The manuscript brings interesting results regarding the weathering process at microbiological scale. By highlighting the role of microbiota in pedogenesis processus. The introduction is well written. However, the references have to be cited following the journal standards, and the main manuscript need be structured following the section suggested by Authors Guide (Introduction, Materials and Methods, Results, Discussion, and Conclusions). The results are well presented and interpreted.
We truly appreciate the valuable suggestions from the reviewer.
The references have been cited following the journal standards now. The section of the manuscript now is structured as suggested.
Find below some specific comments:
Line 85: the author needs to be consistent in using the abbreviation GC-MS.
We appreciate the careful checking from the reviewer. The abbreviation of GC-MS has now been unified throughout the manuscript.
Line 159: add the coordinate of the sampling points.
Thanks to the reviewer’s helpful suggestion. The coordinate for sampling both rocks (37°55’08” N, 4°49’36” W) was added in line 108.
Comments on the Quality of English Language
Minor language correction is needed.
As the reviewer suggested, we re-examined the manuscript with a professional tool ProWritingAid (https://app.prowritingaid.com). Errors in spelling, grammar, and formality have been corrected accordingly.
Round 2
Reviewer 1 Report
Comments and Suggestions for Authors
The authors have made significant changes to the manuscript, I believe that it can be published in this form